# Before Forgetting, There's Learning: Representation Learning Challenges in Online Unsupervised Continual Learning

**Cameron Taylor**                                                           *cameron.taylor@gatech.edu*
*School of Computer Science, Georgia Institute of Technology*

**Shreyas Malakarjun Patil**                                                  *smpatil_7@gatech.edu.com*
*School of Computer Science, Georgia Institute of Technology*

**Constantine Dovrolis**                          *constantine@gatech.edu, c.dovrolis@cyi.ac.cy*
*Schhol of Computer Science, Georgia Institute of Technology*
*Computational Science and Technology Research Center (CaSToRC), The Cyprus Institute*

**Reviewed on OpenReview:** *https://openreview.net/forum?id=hZwInyuYDw*

## Abstract

This paper addresses the Online Continual Unsupervised Learning (O-UCL) problem, where a learner must adapt to a stream of data arriving sequentially from a shifting distribution without storing past data or relying on labels. This challenge mirrors many real-world machine learning applications, where efficient training and updating of large or on device models is critical. We first explore the unique challenges of O-UCL and identify a secondary failure mode in addition to catastrophic forgetting. We demonstrate that the presence of transient, small-scale biases in an online data stream can significantly impair learning. Unlike traditional notions of distribution shift that manifest over long timescales, we highlight how biases occurring at the level of individual batches or short segments—while imperceptible in aggregate—can severely hinder a model's ability to learn, a phenomenon we call "catastrophic non-learning". We further showcase how an auxiliary memory can be used to solve both catastrophic forgetting and catastrophic non-learning, but that the criteria for the ideal memory for each are in conflict. In response to these findings, we introduce a dual-memory framework which incorporates specifically designed modules to mitigate both catastrophic non-learning and forgetting. We validate our findings on challenging, realistic data streams derived from ImageNet and Places365, comparing against multiple baselines to highlight the distinct nature of this problem and the need for new approaches in O-UCL. The code is available on github *here*.

## 1 Introduction

The Continual Learning problem has been widely studied in recent years with many papers approaching the problem from different perspectives Wang et al. (2024). While continual learning works typically focus on understanding and combating catastrophic forgetting, this is not the only challenge that a true "continual" learner would face. In many real-world scenarios, where data is collected directly from the environment, storing and labeling this data may be impractical due to costs or privacy concerns. Even with the vast amounts of potential pre-training data, many potential application domains will have growing and evolving distributions. It would be ideal for these models to grow and expand their understanding as their environments evolve.

Recently, to address some of these challenges the Unsupervised Continual learning (UCL) problem was introduced in Rao et al. (2019). UCL poses the problem of sequentially learning from a series of tasks $\{D_1, D_2, ...D_T\}$. The learner receives one task dataset $D_i$ at a time and can sample batches $b_i = x_{i_1}^B$ such

that $x_i \sim D_i$. Given a candidate learning model $F_\theta(\cdot)$ parameterized by $\theta$, the objective is to learn to extract representations $z = F_\theta(x)$ that are useful for other down-stream tasks such as classification or clustering. Popular approaches for UCL include continually learned generative models Ye & Bors (2020); Achille et al. (2018); Ramapuram et al. (2020); Rao et al. (2019), memory replay Cha et al. (2021); Madaan et al. (2021), distillation Fini et al. (2022); Gomez-Villa et al. (2024); Sadeghi et al. (2024), or both distillation and replay Zhang et al. (2024).

Other recent work has focused on the online version of continual learning known as Online Continual Learning (OCL) Lopez-Paz & Ranzato (2017). OCL poses the problem of learning online from a stream $S(t)$. Examples $(x_i, y_i)$ arrive one at a time, in some cases including a task identifier $t_i$. Each example is seen only once. While these methods address the challenges selecting good memories in an online manner Aljundi et al. (2019a); Guo et al. (2022); Aljundi et al. (2019b); De Lange & Tuytelaars (2021); Chaudhry et al. (2018); Wei et al. (2023), continually learning on top of a pretrained model Harun et al. (2023); Hayes et al. (2019); Hayes & Kanan (2020); Hayes et al. (2020), little focus is given to the challenges of learning online directly.

While the UCL and OCL problems represent important steps in moving the field of continual learning toward more realistic scenarios, we consider an even more challenging scenario where the learning is done both without labels and online. We refer to this problem as Online Unsupervised Continual Learning (O-UCL). O-UCL is inspired by many real-world machine learning applications where a critical challenge is the *training* and *updating* deployed models in an efficient manner. The problem was first introduced in Smith et al. (2019) where it was called Unsupervised Progressive Learning (UPL). Several other papers followed which looked at various aspects of the UPL/O-UCL problem such as improving memory building Yu et al. (2023), better augmentations Michel et al. (2023), or sleep as a form of offline optimization Taylor et al. (2025). While these papers focus on an online and unlabeled version of the CL problem, they still consider learning sequentially from a series of distinct tasks, where the primary challenge is to avoid catastrophic forgetting while learning from a limited amount of data. Another closely related work is Hu et al. (2021a), which investigates pretraining from unsupervised streaming data. While conceptually similar, their approach is oriented toward learning pretrained networks from the stream that are subsequently adapted to downstream tasks. In contrast, our work emphasizes constructing and maintaining useful representations directly throughout the stream itself that are as useful as possible given the stream data seen so far.

Critically, instead of focusing on the "single-pass" setting, we argue that learning from an online stream is instead challenging because of the inherent lack of control over the ordering and sampling of incoming data. In many real-world scenarios, data does not arrive in a perfectly IID manner at small timescales, even if it appears IID when aggregated over longer periods. For instance, a robot navigating an environment may encounter a series of visually similar frames while turning a corner, causing its representation learning to be temporarily biased toward a subset of visual features. Similarly, a recommendation system that updates in real-time may see a burst of interactions from a niche user group, skewing its learned representations before the broader population is observed. In both cases, the learner is not just at risk of forgetting past knowledge but may fail to develop a meaningful representation in the first place. This perspective shifts the focus from solely mitigating catastrophic forgetting to understanding when and why learning itself can fail due to transient biases in the data stream. Our work highlights this fundamental challenge and demonstrates its impact, providing insights into how online learning dynamics differ from conventional continual learning settings.

## 1.1 The O-UCL Problem

We consider an **online representation learning setting** in which an unlabeled data stream arrives sequentially. Rather than assuming a fully non-stationary process, we model the stream as **piecewise-stationary**, meaning that there exist distinct but unknown **phases** during which the data distribution remains stable before shifting unpredictably.

Formally, let $\mathcal{S}(t)$ denote the data stream, where data points $x_t \in \mathcal{X}$ arrive sequentially. We assume that time is partitioned into **phases** $T_1, T_2, \ldots, T_k$, with unknown boundaries. Within each phase $T_k$, the data follows a stationary distribution $P_k(x)$, which is drawn from a finite but unknown set of classes $N_k$. Consecutive phases

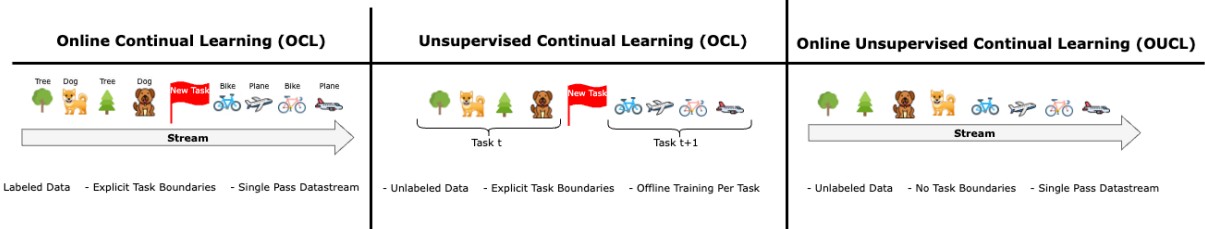

Figure 1: Explanation of the key differences between OCL, UCL, and O-UCL (the setting considered in this work)

may introduce new classes, reintroduce previously seen ones, or shift their relative frequencies, creating an evolving but structured learning environment.

The goal is to learn a **representation function**

$$f_{\theta,k} : \mathcal{X} \to \mathcal{Z}$$

parameterized by $\theta$, that captures salient features of the data encountered up to the current phase $T_k$. Unlike conventional representation learning, where $f_\theta$ is trained on a static dataset, here it must evolve over time, adapting incrementally as new phases arrive.

Specifically, by the end of phase $T_k$, the model should have learned a latent space $\mathcal{Z}_k$ that enables effective downstream tasks—such as clustering or classification—over the union of all classes encountered up to that point, i.e.,

$$N_1 \cup \cdots \cup N_k.$$

At each timestep $t$, the model may incorporate the newly arrived data $x_t$ (or a mini-batch thereof) into its parameters $\theta$, but it does not control the order or composition of the stream. The learner may also store auxiliary information such as aggregate statistics, embeddings, or even a selection of raw images to aid in representation learning. Unlike many continual or online learning problems, we do not impose a strict cap on the total amount of storage; however, we emphasize that memory usage should be **efficient**, ensuring that the storage requirements remain manageable as the stream progresses. The challenge is not only to determine **what** to store but also **how** to leverage this auxiliary memory to maintain robust and generalizable representations while adapting to distribution shifts over time.

At any time $t$, the model's performance is assessed using **classification accuracy** and **clustering purity** over an evaluation set $\mathcal{D}_k$, which contains data sampled from $N_1 \cup \cdots \cup N_k$. The challenge is not only to avoid **catastrophic forgetting** of past knowledge but also to ensure that learning remains stable in the presence of small, local biases within individual phases—biases that may not be apparent when viewing the data distribution of the current phase in aggregate.

## 2 Challenges of Online Unsupervised Continual Learning

The O-UCL problem introduces multiple challenges for a learner, but we primarily focus on two key aspects: (1) learning online from poorly mixed, highly biased data streams, and (2) maintaining memory-efficient representations while avoiding forgetting, despite the absence of class or phase labels. To systematically investigate these challenges, we conduct a series of experiments using a consistent model architecture, training objective, and hyperparameter configuration across all settings.

In each of the following subsections, we utilize a neural network $F_\theta(\cdot)$ as our core learning model. Specifically, $F_\theta(\cdot)$ is a **ResNet-18 backbone with a projection head** trained using the SimCLR Chen et al. (2020a)

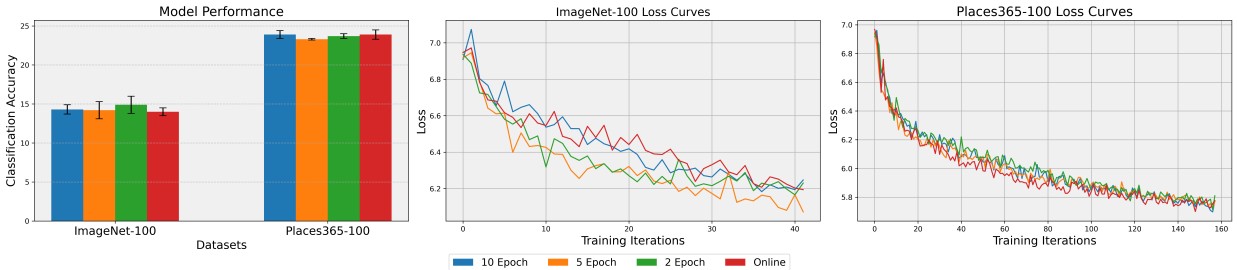

Figure 2: KNN-Accuracy of models (left) trained with varying amounts of data for a fixed number of iterations. The four models were trained with 10% of the data for 10 epochs, 25% for the data for 4 epochs, 50% of the data for 2 epochs, and 100% of the data for 1 epoch (online). Additionally, loss curves for a single seed per model on ImageNet-100 (middle) and Places365-100 (right). The results represent the mean and $\pm$ one standard deviation over three independently seeded trials for both ImageNet-100 and Places365-100.

self-supervised learning framework. The training objective follows a contrastive loss formulation, where positive pairs are constructed using the augmentations from Chen et al. (2020a). While we adopt SimCLR as our primary learning paradigm, our framework is agnostic to the specific self-supervised objective, and alternative contrastive or non-contrastive approaches should also be applicable. The model is optimized using **SGD with a fixed learning rate of 0.6**, a batch size of **512** and temperature of **0.5**. All models are trained for the equivalent of 100 epochs in terms of total training iterations.

For each experiment, the total number of classes is 100 from either ImageNet (ImageNet-100) or Places365 (Places365-100). Each experiment is done with a train, validation, and test split of 80%, 10%, and 10% respectively. All results are presented as the mean over three independently seeded trials, with error measurements represented as plus or minus one standard deviation. Further details on the construction of phases and stream sampling biases vary by subsection and are discussed individually in each.

## 2.1 Online Learning as a "Single-Pass"

A common assumption in online learning research is that each data point should only be seen once, often referred to as the "single-pass" criterion Lopez-Paz & Ranzato (2017); Hu et al. (2021b); Taylor et al. (2025); Yu et al. (2023). This restriction significantly reduces the number of training iterations and forces the model to extract meaningful representations in a highly constrained setting. However, it remains unclear whether this assumption is fundamental to online learning or whether the assumption that a model cannot revisit any examples "on-demand" is a sufficient criteria for simulating a realisitic learning scenario.

We seek to understand:

> *Do we need to have the repetition of examples in multiple epochs in order to learn well?*

To investigate this, we compare several learners under controlled conditions. One learner is trained "online" and visits a total of $N$ examples exactly once each. This is equivalent to training for a single epoch or equivalently for a "single-pass". We compare this with several learners which observe $N_e$ unique examples repeated for $E$ epochs (i.e., revisiting each example $E$ times). For these comparison learners, we keep the total number of training iterations fixed and vary $E$ and $N_e$ such that $E * N_e = N$. All learners observe the same total number of training batches. If revisiting is necessary, it suggests that online learning requires multiple exposures to the same samples for effective training. Conversely, if performance is equivalent, it challenges the notion that single-pass (defined as only seeing each example exactly once) learning is a defining constraint of online learning.

In Figure 2 we showcase the performance of the learners described above. The "online" learners performance, represented by the red bars is compared with other learners which train for $E = 2, 4, 10$ epochs, while seeing $N_E = \frac{N}{2}, \frac{N}{4}$, and $\frac{N}{10}$ of the examples respectively. The results show that there is no statistically significant difference between any of the learners. Additionally, while the loss curves show that none of the models have

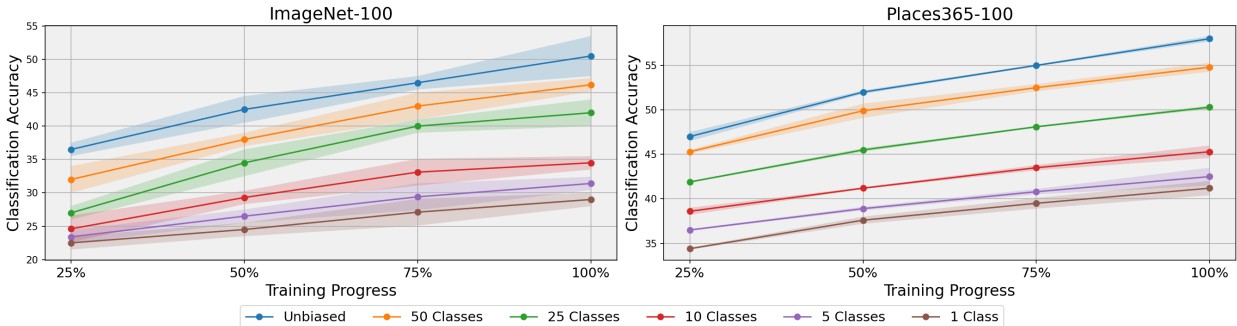

Figure 3: KNN accuracy comparing several learners trained under varying degrees of biased streams. Results are shown for 100-class subsets of ImageNet and Places365. Each result represents the mean and standard deviation over six trials consisting of 3 dataset samples and two random seeds.

"converged" to a nearly constant loss value, they follow very similar paths further indicating that despite the number of unique data points $N_E$, being very different, the dominant factor in performance is the total number of training iterations.

Based on these results, we argue the criteria that the learner sees each image only once should not be a requirement of an online learning scenario, but instead the requirement should be that it cannot store all the images and cannot control *when* or *if* an image is seen again. This allows for the creation of significantly longer streams, created from academic datasets, granting us more flexibility to understand other unique challenges in online learning such as the one presented in the next subsection.

## 2.2 A New Challenge in Online Learning

An intrinsic difficulty in online learning from data streams is that, even when the long-term distribution of the stream is stationary, the short-term composition of samples may exhibit significant deviations. In practice, data streams are rarely well-mixed; there are often temporal correlations or local biases that cause certain subsets of the data distribution to be overrepresented for short periods.

For example, consider a robotic agent navigating an environment. If the agent moves through a narrow corridor, its observations may be dominated by a limited set of objects or textures, biasing its short-term experience. Similarly, in an autonomous driving scenario, consecutive frames may contain nearly identical visual features due to a temporarily static viewpoint. While these biases may dissipate over longer timescales, they can significantly impact the learning dynamics when training occurs in an online manner. Unlike long-term distribution shifts, which have been extensively studied in the continual learning literature Wang et al. (2024), these short-term biases introduce a different challenge: they may distort the loss landscape for each batch. This is because biased batches skew gradient directions away from minimizers of the long-term distribution, effectively reshaping the loss landscape locally. This makes optimization trajectories diverge from globally useful representations, making it difficult for the learner to find a solution that generalizes to the full data distribution.

We explore this by investigating:

> *How do short-term or local biases in the stream sampling process impact learning?*

To quantify the effect of short-term biases in stream samples, we conduct an experiment where we introduce varying degrees of local sampling bias. We consider a stream $S(t)$ consisting of $K$ unique classes and control the degree of bias by restricting how many classes appear within each batch over a short window of time. Specifically, instead of sampling batches uniformly from the full distribution, we constrain each batch to be drawn from a subset of classes, where the size of this subset varies across conditions.

We compare six different learners, each trained under a different degree of short-term bias:

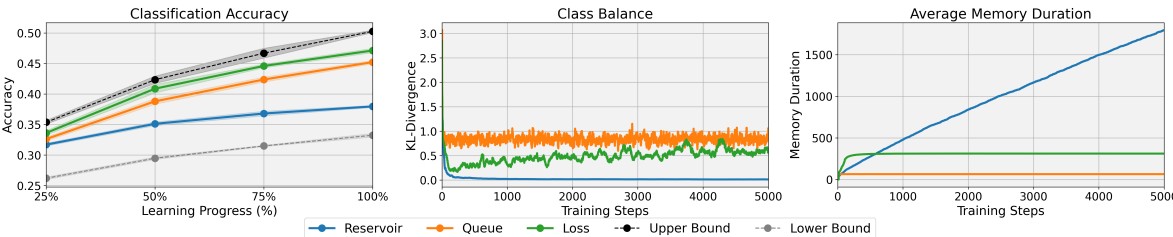

Figure 4: Classification accuracy (left), class balance (center), and average memory duration (right) for models utilizing memory buffers with different selection and removal strategies. The training stream is constructed from 100 classes of ImageNet with a bias of one class per batch. Classification results represent the mean and standard deviation of three independent trials, while class balance and average memory duration are computed from a single trial.

1. **Unbiased baseline**: Each batch is sampled IID from all 100 classes, ensuring well-mixed training data.

2. **50 class bias**: Each batch is sampled from a subset of 50 randomly selected classes.

3. **25 class bias**: Each batch is sampled from a subset of 25 randomly selected classes.

4. **10 class bias**: Each batch is sampled from a subset of 10 randomly selected classes.

5. **5 class bias**: Each batch is sampled from a subset of 5 classes.

6. **1 class bias**: Each batch is sampled from only a single class at a time.

The impact of these biases is evaluated on two datasets, ImageNet and Places365.

Figure 3 presents the results of this experiment. As expected, the model trained with unbiased samples (blue curve) achieves the highest performance. However, even mild levels of bias (orange and green curves) result in noticeable degradation in downstream classification performance. With more severe bias (red, purple, and brown curves), performance deteriorates dramatically, confirming that local sampling biases significantly hinder representation learning.

Unlike catastrophic forgetting, where knowledge of past distributions is lost, these short-term biases introduce a different failure mode: catastrophic non-learning. Since the learner takes gradient steps based on a distribution that is not representative of the long-term data, it struggles to construct a meaningful representation that generalizes beyond its immediate experience. This highlights the need for approaches that mitigate the effect of short-term biases when learning from non-randomly ordered data streams. It may also be possible that this problem is unique to the SimCLR approach and might be mitigated by another SSL approach such as MoCo Chen et al. (2020b) which utilizes a memory bank. We explore this possibility and demonstrate why this is not the case in Appendix E.

## 2.3 A Memory Buffer for Biased Streams

As discussed in the previous subsection, the learner has no control over the stream sampling process, meaning that each batch may be a poor representation of the overall distribution. This results in biased gradient estimates, which can hinder learning. One way to mitigate this issue is by maintaining a small memory buffer to store past examples, enabling the model to construct more representative batches. However, the effectiveness of a memory buffer may depend on factors such as *which* examples are stored and for *how long*.

We seek to understand:

*What are the important criteria for designing an effective short-term bias mitigating memory buffer?*

To answer this, we consider two criteria for memory selection: 1) **Class balance**—ensuring that the buffer maintains a diverse set of examples across classes. 2) **Example duration**—controlling how long individual examples persist in memory to prevent overfitting. An ideal memory buffer should maximize class balance, minimize example duration, and prioritize high-loss contributing examples to enhance learning efficiency.

We evaluate the impact of different memory selection strategies by comparing three buffer implementations along with a lower and upper bound:

- **Lower Bound**: Trains from a biased stream without any auxiliary memory buffer.

- **Reservoir Sampling**: Maintains a uniform sample of past data, ensuring class balance but reducing throughput and learning value.

- **FIFO Queue**: Stores all incoming examples in a first-in, first-out manner, maximizing throughput but potentially sacrificing class balance and learning value.

- **Loss Buffer**: Selects a subset of high-loss examples from each batch, aiming to balance throughput and class diversity.

- **Upper Bound**: Trains from an unbiased stream without any auxiliary memory buffer.

To quantify the effectiveness of each buffer, we measure *classification accuracy*, *class balance* (via KL divergence between buffer class distribution and a uniform distribution), and *example duration* (average time an example remains in memory).

As shown in Figure 4, all memory buffers significantly improve performance compared to training without memory. The FIFO and Loss Buffers achieve similar accuracy, whereas the Reservoir Buffer underperforms, which we attribute to its higher average memory duration. This suggests that while class balance is important, excessive memory persistence can negatively impact learning by reducing exposure to recent data.

Additionally, the Loss Buffer slightly outperforms the FIFO Queue, likely due to its improved class balance and selective retention of challenging examples. These findings indicate that the optimal buffer must balance short-term diversity with retention of informative samples.

## 2.4 From Local Biases to Catastrophic Forgetting: The Impact of Multi-Phase Streams

The previous section examined the impact of local sampling biases in a stream where all classes remained present throughout training. While such biases hinder representation learning, they do not introduce *permanent* shifts in the data distribution. However, in many real-world settings, the set of observed categories changes over time, requiring the learner to adapt to new data while retaining past knowledge.

In this subsection, we transition from a single-phase stream (100 classes throughout) to a multi-phase stream (five phases of 20 classes each), which introduces *catastrophic forgetting* Wang et al. (2024). Unlike *catastrophic non-learning*, where biased sampling prevents effective learning, catastrophic forgetting occurs when past representations are overwritten by newer information. While we showed in the previous subsection that buffers with relatively high throughput are the most effective for dealing with the short-term biases, this will likely not be effective for avoiding the long-term shifts which introduce catastrophic forgetting.

We now investigate:

*How well does a buffer designed for avoiding catastrophic non-learning also mitigate catastrophic forgetting?*

To study this, we modify the previous stream by partitioning it into five sequential *phases*, each containing a disjoint set of 20 classes. Once a phase ends, its classes are never seen again. In this subsection only, we consider a phase-wise evaluation (similar to task-incremental evaluations in traditional continual learning). Phase-wise accuracy for phase $T_i$ is measured immediately after training on it, denoted as $A_{i,i}$, while final accuracy on $T_i$ after completing all $K$ tasks is $A_{i,K}$. Phase-wise forgetting is quantified as the difference between the initial and final accuracy on each task, defined as $F_i = A_{i,i} - A_{i,K}$.

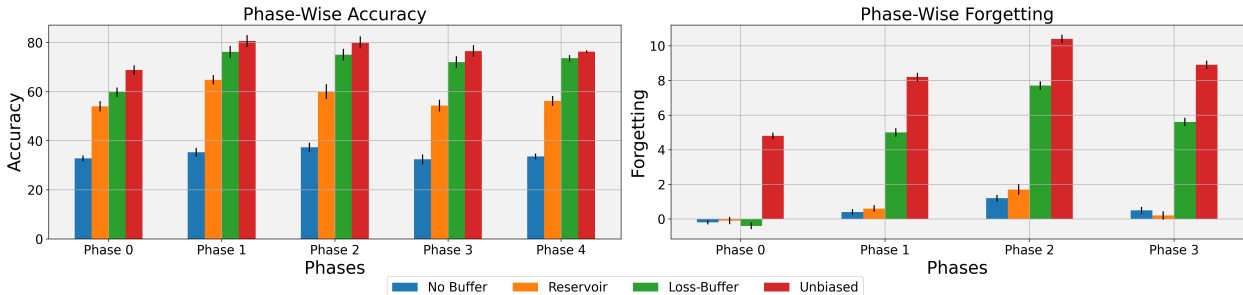

Figure 5: Phase-wise classification accuracy and forgetting for several memory buffer implementations. The stream consists of 100 classes from ImageNet, split into five phases of 20 classes each. Performance is measured at the end of each phase. Results represent the mean and standard deviation over three independently seeded trials.

We evaluate two memory strategies along with a lower and upper bound:

- **No-Mem Lower Bound**: Learns with no memory or buffer to improve learning or avoid forgetting.

- **Reservoir Sampling**: Maintains a uniformly sampled memory to improve retention.

- **Loss Buffer**: Prioritizes high-throughput learning by frequently updating stored examples.

- **Unbiased Upper Bound**: Learns from an unbiased stream, that still consists of 5 total phases.

Unlike previous experiments measuring overall accuracy, we now track *phase-wise accuracy* and *phase-wise forgetting*. Accuracy is evaluated at the end of each phase, while forgetting is measured as the difference between peak and final accuracy for each phase.

Figure 5 presents the results. The Loss Buffer achieves strong accuracy during training, closely matching the performance of an unbiased upper-bound. However, it suffers significantly higher forgetting compared to Reservoir Sampling.

This highlights a key trade-off: high-throughput learning optimizes initial performance but increases forgetting. The Loss Buffer rapidly adapts to new phases but at the cost of overwriting older representations, whereas Reservoir Sampling, while less efficient in short-term learning, better preserves past knowledge. In the next subsection, we explore the important design criteria of a memory for avoiding forgetting, that must select examples online and without labels or phase boundary information.

## 2.5 A Memory Buffer for Avoiding Forgetting

In most existing work, a memory or buffer would be assumed to have a fixed capacity. However, because we consider streams which may have extremely long or even *infinite* length, we argue that a more appropriate criteria is to focus on growth efficiency with respect to the number of novel concepts or objects introduced. We propose that an effective replay memory must achieve *phase-balance*—storing new information when distribution shifts occur—while remaining *efficient*, preventing uncontrolled or unnecessary growth.

The key challenge is that, without class labels or phase boundaries, the learner must autonomously determine which examples to retain. This is further complicated by the evolving latent space $\mathcal{Z}(t)$, where feature distances and similarity relationships shift over time.

We explore this by asking:

*How can we design a memory which maintains a balanced and efficient example set under these conditions?*

To explore this, we examine a similarity-based memory that selectively expands when novel examples appear and periodically consolidates redundant entries. Specifically, new samples are compared against stored ex-

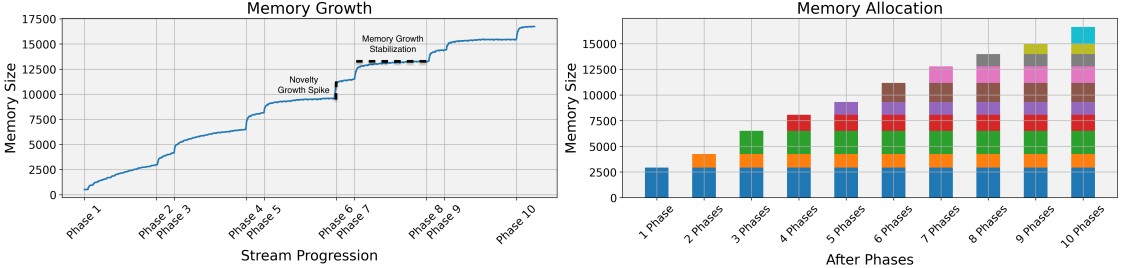

Figure 6: Memory growth and per-phase storage distribution for a similarity-based memory. The stream consists of 100 classes from ImageNet partitioned into ten phases. Even-numbered phases are four times longer, creating significant class and phase imbalance. The memory growth resembles a sort of "step function" where a spike in growth is observed after the introduction of novel data at the beginning of each new phase and then a period of relative stability and minimal growth is reached until a new phase is introduced.

amples using cosine similarity. If an example is sufficiently dissimilar (beyond an adaptive novelty threshold), it is added to the memory. To prevent uncontrolled growth, a consolidation step periodically prunes highly redundant examples as the latent space evolves.

To evaluate this mechanism, we track memory growth in a data stream with no class labels or phase boundaries, where the distribution shifts progressively. We measure: 1) **Memory size over time**—does the buffer grow indefinitely or stabilize? 2) **Storage distribution across phases**—does the buffer allocate memory equitably despite imbalanced exposure?

As shown in Figure 6, our similarity-based memory successfully expands when novel information appears (annotated as "novelty growth spike") and stabilizes in size over time (annotated as "memory growth stabilization"). Despite significant phase imbalances, memory allocation per phase remains close to uniform, indicating that the buffer reacts to distribution shifts rather than raw frequency. Importantly, some phases require more storage due to increased novelty, preventing a strictly uniform allocation. However, it is important to note that these variances in storage allocation are *not* correlated with the length of the phase.

Moreover, memory growth trends slightly downward as the stream progresses, suggesting that earlier features remain useful, reducing the need for continuous expansion. These results highlight that, even in the absence of explicit labels or phase information, a well-designed similarity-based memory can maintain a compact yet adaptive replay buffer. Something which, to the authors knowledge, has not been demonstrated previously.

## 3 A Dual-Memory Architecture for Online Unsupervised Continual Learning

The results from Section 2.3 demonstrated that in an online setting, short-term biases in batch sampling can significantly impair learning. In Section 2.4, we showed that mitigating catastrophic forgetting requires a different memory strategy what is useful for online learning. While an ideal memory would balance short- and long-term biases by maintaining a diverse set of examples, the two differ fundamentally in their effective throughput. Relying solely on short-term memory leads to catastrophic forgetting, as older examples are continually overwritten and never permanently stored. Conversely, relying exclusively on long-term memory results in poor overall performance, since the model primarily learns from a relatively static and limited subset of examples. Motivated by these findings, we propose a *dual-memory architecture* that combines two memory modules to accommodate online learning from a non-uniform stream:

- A **Short-Term Memory (STM)** designed to maintain high-throughput and balanced batch composition for improving online learning efficiency.

- A **Long-Term Memory (LTM)** designed to expand dynamically while maintaining a compact set of diverse and representative examples to prevent forgetting.

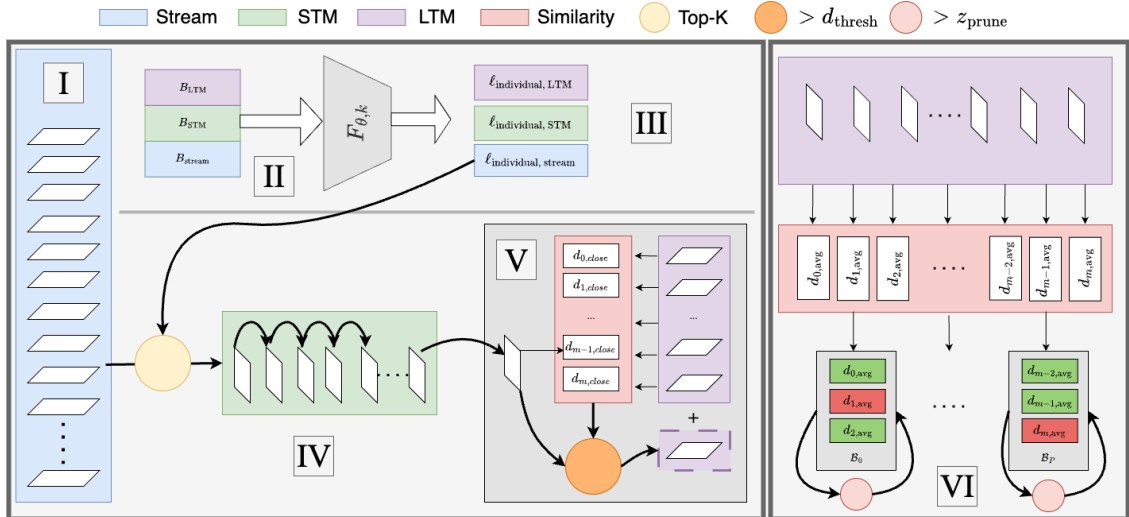

Figure 7: I) Images arrive one at a time as a stream. II) Batch is sampled from $B_s$, $B_{\text{STM}}$, and $B_{\text{LTM}}$ and fed into the encoder $F_{\theta,k}$. III) The individual losses are collected and those coming from stream examples are passed to the STM. IV) The STM collects the top K examples from $B_S$, evicting the $K$ oldest examples. V) Recently evicted STM examples are compared with existing LTM examples, if the examples are dissimilar enough ($> d_{\text{thresh}}$), they are added to the LTM. VI) Examples are partitioned into $P$ partitions based on age and examples, which are outliers within their respective partition (z score $> z_{\text{prune}}$), based on high similarity with their closest neighbor, are pruned.

This framework provides a principled way to balance learning efficiency and memory retention in the online unsupervised continual learning (O-UCL) setting. We also emphasize that while we provide specific implementations for the STM and LTM, the contributions of this paper focus on the need for a dual-memory system based on the conflicting criteria necessary to design an appropriate STM and LTM.

### 3.1 Short-Term Memory (STM)

The STM is responsible for mitigating the adverse effects of short-term biases in the data stream. Since incoming batches may be highly imbalanced, the STM must (1) sample examples that maximize learning utility and (2) ensure that examples are not overrepresented in future batches. As demonstrated in Subsection 2.3, these requirements suggest two key criteria:

- **High class diversity**: The buffer should contain examples from as many different classes as possible to prevent a dominant subset from biasing training.

- **Low retention duration**: Examples should be replaced frequently to maximize exposure to new samples.

**STM Selection Mechanism**    At each time step $t$, the STM maintains a buffer $\mathcal{B}_{\text{stm},t}$ of fixed size $M_{\text{stm}}$. To populate this buffer, we select the top $K$ examples from the incoming batch $X_t$ that have the highest per-example loss, which serves as a proxy for their learning utility.

$$X_{\text{stm},t} = \text{argmax}_{X' \subset X_t, |X'|=K} \sum_{x \in X'} \ell_{\text{individual}}(F_\theta(x)). \tag{1}$$

Once the STM reaches capacity, the oldest examples are evicted in a first-in-first-out (FIFO) manner. These evicted examples are then passed to the LTM for potential long-term storage.

**Individual InfoNCE Loss** Following the standard SimCLR framework Chen et al. (2020a), each input $x$ is augmented into two different views, $x_i$ and $x_j$, and passed through an encoder $F_\theta$ followed by a projection head $g_\theta$, yielding the embeddings:

$$z_i = g_\theta(F_\theta(x_i)), \quad z_j = g_\theta(F_\theta(x_j)). \tag{2}$$

The contrastive loss for a single example is computed as:

$$\ell_{\text{InfoNCE}}(x_i, x_j) = -\log \frac{\exp(\text{sim}(z_i, z_j)/\tau)}{\sum_{x_k \in X_t, k \neq i} \exp(\text{sim}(z_i, z_k)/\tau)}, \tag{3}$$

where $\text{sim}(z_i, z_j) = \frac{z_i \cdot z_j}{\|z_i\|\|z_j\|}$ is the cosine similarity, and $\tau$ is the temperature parameter. The individual loss for an example is then defined as the average loss across both views:

$$\ell_{\text{individual}}(F_\theta(x)) = \frac{1}{2}\Big(\ell_{\text{InfoNCE}}(x_i, x_j) + \ell_{\text{InfoNCE}}(x_j, x_i)\Big). \tag{4}$$

### 3.2 Long-Term Memory (LTM)

While the STM helps improve online learning, it does not address long-term retention. The LTM must (1) ensure maintenance of a representative set of examples and 2) do so in an efficient manner (preventing uncontrolled memory growth). As demonstrated in Subsection 2.5 these requirements suggest two key criteria:

1. **Balanced Storage Allocation**: The memory should maintain a balanced set of examples with respect to all classes or objects seen so far.

2. **Memory Efficiency**: The memory should be efficient in its selection of examples, expanding only when novel data is encountered.

**LTM Selection Mechanism** To achieve this, we rely on a similarity-based selection strategy that dynamically expands storage based on novelty. The LTM stores a growing set of embeddings $Z_{\text{ltm}}$ derived from past data. At each time step, evicted STM examples are compared against existing LTM examples using cosine similarity:

$$D = 1 - (Z_{\text{stm}} Z_{\text{ltm}}^T). \tag{5}$$

For each new example embedding $z_i$, we compute its nearest neighbor distance in the LTM:

$$d_{i,\text{close}} = \min_{j \in \mathcal{B}_{\text{ltm}}} D_{i,j}. \tag{6}$$

If $d_{i,\text{close}} > d_{\text{thresh}}$, the example is added to memory. The threshold $d_{\text{thresh}}$ is updated dynamically based on the $\beta$-percentile of nearest neighbor distances in the current memory:

$$d_{\text{thresh}} = P_\beta(\{d_{i,\text{close}} \mid i \in \mathcal{B}_{\text{ltm}}\}). \tag{7}$$

To reduce computational overhead, we maintain a frozen copy of the encoder $F_\theta(x)$, updating it every $u$ steps to ensure stability and efficiency. This step is critical for achieving the growth pattern we observe in Subsection 2.5 and also allows the computational requirements to be significantly reduced, by avoiding the need to recalculate all $Z_{\text{ltm}}$ each step.

**Memory Consolidation**  As training progresses, some stored examples may become redundant due to the evolving feature space. To maintain efficiency, we periodically prune redundant examples based on their average pairwise cosine distance:

$$d_{i,\mathrm{avg}} = \frac{1}{M_{\mathrm{ltm}} - 1} \sum_{j \neq i} D_{i,j}. \tag{8}$$

However, since embeddings introduced earlier in training tend to have higher pairwise distances than more recent embeddings, pruning based on global statistics can lead to an undesirable bias toward removing older examples. To account for this, we partition the memory into $P$ chronological bins based on insertion time, such that each partition contains examples from a distinct time period:

$$\mathcal{B}_{\mathrm{ltm}} = \bigcup_{p=1}^{P} \mathcal{B}_p, \quad \text{where} \quad \mathcal{B}_p = \{x_i \in \mathcal{B}_{\mathrm{ltm}} \mid t_i \in \Gamma_p\}. \tag{9}$$

Here, $\Gamma_p$ represents the time range for partition $p$, ensuring that each partition contains examples added to memory during a specific interval. This helps to reduce any bias towards different phases or classes that may occur based on the duration with which they have been observed.

For each partition, we compute the mean $\mu_p$ and standard deviation $\sigma_p$ of the pairwise distances and use this to compute the z-score for each example within its parition. An example is pruned if its Z-score falls below a predefined threshold $z_{\mathrm{prune}}$:

$$\mathcal{B}_p \leftarrow \mathcal{B}_p \setminus \{x_i \mid z_i < z_{\mathrm{prune}}\}. \tag{10}$$

By partitioning based on insertion time and performing per-partition pruning, we prevent systematic bias against newer, relatively more similar, memory entries and ensure that only truly redundant examples are removed.

### 3.3   Batch Sampling Strategy

At each training step, we construct a batch $X_t$ of size $B$ from the stream, STM, and LTM:

$$B_s + B_{\mathrm{stm}} + B_{\mathrm{ltm}} = B. \tag{11}$$

We allocate a fixed fraction ($B_s = 12.5\%B$) to stream samples to maintain exposure to new data. The remaining batch is split dynamically between STM and LTM, weighted by their relative memory sizes:

$$B_{\mathrm{stm}} = \frac{\mathcal{M}_{\mathrm{stm}}}{\mathcal{M}} B_M, \quad B_{\mathrm{ltm}} = \frac{\mathcal{M}_{\mathrm{ltm}}}{\mathcal{M}} B_M, \quad B_M = B - B_s. \tag{12}$$

This ensures that early training relies more on STM (where forgetting is less of a concern), while later training shifts towards LTM as memory accumulates.

This framework ensures that online learning benefits from a high-throughput STM for fast adaptation while maintaining a dynamic LTM for long-term retention.

## 4   Experimental Results

In this section, we evaluate the proposed dual-memory system in terms of representation quality, memory efficiency, and adaptability to distribution shifts in an online unsupervised setting without explicit labels.

### 4.1 Experimental Details

**Training Details**  All models use a ResNet-18 backbone. Training follows the SimCLR self-supervised approach Chen et al. (2020a), including data augmentations and the InfoNCE loss. Optimization is performed using SGD with a learning rate of 0.6, momentum of 0.9, and weight decay of $1 \times 10^{-6}$. Downstream classification is evaluated using K-nearest neighbors (KNN) with $K = 50$, while clustering is assessed via K-means, with $K = 2 * N_T$ applied to the learned embeddings.

**Stream Construction**  We evaluate models on data streams constructed from ImageNet Deng et al. (2009) and Places365 Zhou et al. (2017), each containing 100 randomly selected classes. To introduce batch-level bias, each batch is sampled from a single class randomly drawn from the current phase. Streams are divided into 10 sequential phases of 10 classes each (except where otherwise noted), with alternating phase durations to induce an imbalance: even-numbered phases span the equivalent of 400 epochs (batch size 512), while odd-numbered phases span 100 epochs. The dataset is split into 80% for training, 10% for supervision (used to train KNN), and 10% for testing. For both datasets we resize the images to 128x128 and apply standard normalization.

**Evaluation Metrics**  Models are evaluated on two downstream tasks: classification and clustering. Classification accuracy is measured after each phase, as well as averaged over all evaluations. Clustering performance is quantified using cluster purity. We adopt a class-incremental learning evaluation protocol Zhou et al. (2024), where the classification and clustering tasks expand to include all previously seen classes, without phase labels during training or evaluation. Since performance degradation may result from either increased class ambiguity or catastrophic forgetting, we compare against baselines (detailed below) that provide reference points for expected performance.

### 4.2 Baselines

Due to the uniqueness of the proposed approach, existing methods such as STAM Smith et al. (2019), PCMC Taylor et al. (2025), and SCALE Yu et al. (2023) are not directly applicable. STAM was designed for simpler datasets (e.g., MNIST, SVHN) and struggles with complex images Taylor et al. (2025). PCMC relies on an offline sleep phase, making it unsuitable for our continuous learning setting. SCALE employs memory clustering at each step, resulting in quadratic complexity with respect to memory size, rendering it infeasible for long streams. Other potential baselines memory approaches in the CL literature such as Yoon et al. (2021); Rebuffi et al. (2017); Tiwari et al. (2022) would also be difficult to adapt to our setting due to their reliance on labeled examples, task boundaries or offline calculations between tasks.

Instead, we evaluate five baselines to establish performance bounds:

**Supervised Offline**  A ResNet-18 is trained on the full distribution of 100 classes in an offline supervised setting using cross-entropy loss. The model is trained for 100 epochs with SGD (lr = 0.1, momentum 0.9, weight decay $1 \times 10^{-4}$), ensuring the same total number of training iterations as the streaming setting. This serves as an upper bound, illustrating performance achievable under ideal conditions with labeled data and no sampling biases.

**Self-Supervised Offline**  An encoder is trained on the full dataset in an offline manner using the SimCLR objective. Training follows the same procedure as the supervised baseline but without class labels and without phase-level biases. This baseline quantifies the performance gap between our model and an offline-trained self-supervised learner, under equivalent training conditions.

**LTM Only**  An encoder trained under the default streaming setting but using only the long-term memory module. The LTM is implemented as the similarity memory described in Section 3.2. The LTM is modified to take in all streaming batch examples instead of those evicted from the STM. This baseline isolates the performance of the model excluding the STM, providing insight into the extent to which the STM assists in learning on top of the LTM which mitigates forgetting.

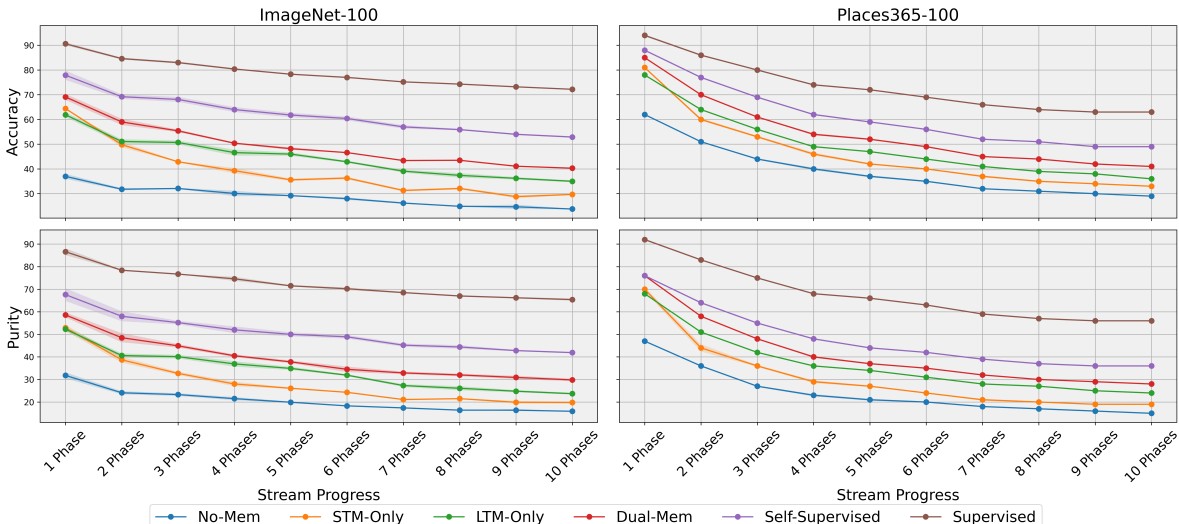

Figure 8: Classification and clustering performance per phase comparing our proposed dual-memory approach with several baselines meant to serve as upper and lower bounds. The top rows represents classification accuracy and the bottom row represents cluster purity. Each data point represents the mean over three independently seeded trials with error represented as ± one standard deviation.

**STM Only**   An encoder trained under the default streaming setting but using only the short-term memory module. The STM is implemented as the loss-buffer described in Section 3.1. This baseline isolates the impact of excluding the LTM, providing insight into the extent of forgetting over time. Performance is expected to initially match the full model but degrade as the number of phases increases due to cumulative forgetting.

**No Memory**   An encoder trained without any memory module. This baseline represents the lower bound on performance, highlighting the benefits of incorporating memory-based replay mechanisms in the proposed approach.

### 4.3   Primary Results

We evaluate the performance of our dual-memory system on a phase-imbalanced stream, comparing it with several baselines. Figure 8 presents classification and clustering accuracy across phases.

**Understanding the Upper Bounds**   The supervised and self-supervised offline baselines provide reference points for how challenging the classification and clustering tasks become as the number of classes increases, despite no changes in the models themselves. Even in these offline settings, performance declines as more classes are introduced, highlighting the inherent difficulty of class-incremental learning. The performance drop from Phase 1 to Phase 10 for the self-supervised model is 24.9%, while for the dual-memory model, it is 28.7%, suggesting that a significant portion of the decline is due to task complexity rather than forgetting. The additional 3.8% degradation in our model reflects an estimate of the impact of catastrophic forgetting.

**Forgetting and Online Learning Challenges**   Comparing the dual-memory system to the STM-only, LTM-only, and no-memory baselines illustrates the necessity of both short-term and long-term memory. The no-memory model exhibits extremely poor initial performance, which deteriorates further due to both learning failures and catastrophic forgetting. The STM-only model starts with much higher accuracy, demonstrating its ability to mitigate early-stage learning failures. However, its performance steadily declines and converges near the no-memory baseline, confirming that STM alone is insufficient for long-term retention. Additionally, the LTM model struggles early due to the biased streams, however as the stream goes on its performance stabilizes. However, it's overall performance is hindered by learning from a stagnant (low-

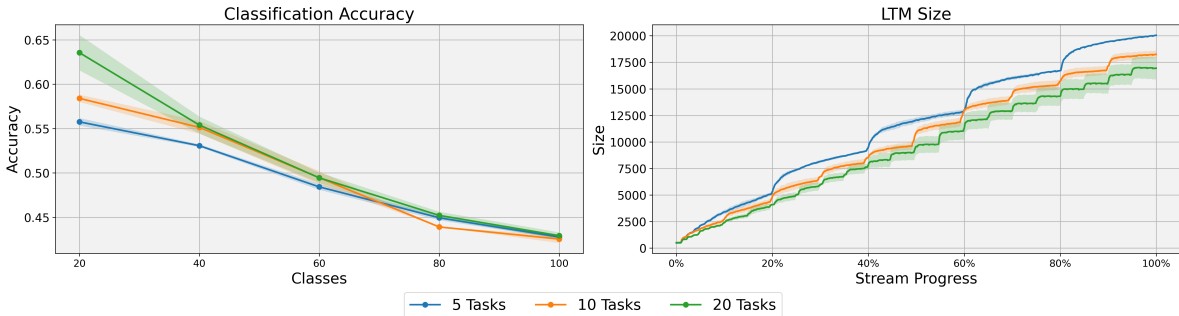

Figure 9: Classification accuracy and LTM growth throughout the stream for the dual-memory system trained on streams of varying task granularity (5, 10, 20) tasks. Each stream is created with alternating task lengths, but with varying total number of classes per task and total tasks.

throughput) pool of examples present in only the LTM. In contrast, the dual-memory approach maintains significantly higher accuracy, with only a modest decline, demonstrating its ability to prevent both learning failures and forgetting.

**Dataset-Specific Trends**  The observed trends hold across both classification and clustering tasks for ImageNet and Places365. However, performance degradation is more pronounced in Places365 despite its higher initial accuracy. This is likely due to its higher per-class data availability but lower inter-class variance, making early phases easier while increasing ambiguity in later phases.

## 4.4   Adaptability to Varying Stream Granularity

A key requirement for the LTM is its robustness to different stream conditions. Figure 9 examines the impact of phase granularity by varying the number of phases while keeping the total number and ordering of classes fixed. This results in streams where the model experiences different levels of class diversity and varying durations of stable distributions, without any changes to hyper-parameters.

**Impact on Classification Performance**  The left subfigure of Figure 9 shows classification accuracy after every 20 newly introduced classes. In the five-phase stream, this corresponds to evaluations at phase boundaries, whereas in the twenty-phase stream, evaluations occur every four phases—after the same number of classes has been observed. Despite initial differences, all configurations exhibit a similar trend, with the 20-phase stream achieving slightly higher early performance, suggesting that finer-grained updates improve initial learning. However, all streams converge to similar accuracy by the end, demonstrating that the dual-memory system remains effective across varying phase granularities.

**Impact on Long-Term Memory Growth**  The right subfigure of Figure 9 illustrates the LTM's growth over time. While the overall trends remain consistent, streams with fewer phases exhibit a faster growth rate. This effect arises due to the similarity threshold $d_{\mathrm{thresh}}$ being slightly larger in finer-grained streams, leading to more selective memory expansion. We suspect this may be caused by quicker convergence for simpler class sets presented in the smaller phases. Despite these differences, final performance remains unchanged, indicating that the LTM dynamically adjusts its capacity based on stream conditions without degrading accuracy.

**Practical Implications**  These results suggest that if the stream's phase granularity is known beforehand, a larger value of $\beta$ for lower phase granularity streams may be beneficial. However, the dual-memory approach demonstrates strong adaptability, ensuring reliable performance across a broad range of phase granularities without requiring hyper-parameter tuning.

### 4.5 Impact of Memory Consolidation Sensitivity

Table 1 examines the effect of varying the LTM consolidation sensitivity, controlled by the pruning threshold $z_{\mathrm{prune}}$. We report final classification accuracy, clustering purity, and final LTM memory size for different values of $z_{\mathrm{prune}}$ on both ImageNet-100 and Places365-100 streams. The setting $z_{\mathrm{prune}} = \infty$ represents a model with no consolidation.

**Trade-Off Between Memory Efficiency and Performance**  Across all settings, reducing $z_{\mathrm{prune}}$ significantly decreases total memory usage, demonstrating the effectiveness of consolidation in preventing unbounded growth. However, for moderate pruning thresholds ($z_{\mathrm{prune}} \geq 2.0$), performance remains nearly unchanged, indicating that many pruned examples were redundant. When pruning is more aggressive ($z_{\mathrm{prune}} = 1.75$), a slight drop in accuracy and purity is observed, suggesting that excessive consolidation may remove informative examples.

**Long-Term Memory Growth Stability**  Although total memory usage at $z_{\mathrm{prune}} = \infty$ is higher than other configurations without consolidation, we observed a more interesting trend in the growth rate of the memory. Specifically, without any pruning the size of the memory continues to grow slowly, despite a period of fixed distribution. This is caused by the definition of novelty continuing to shift over time. By adding consolidation, the LTM is able to reach a steady state where the number of additional "novel" examples is equal to the number of pruned examples. This leads to a slowly improving LTM example set, while keeping the total size fixed.

**Practical Considerations**  These results highlight the importance of consolidation in achieving a stable, bounded memory size while maintaining high performance. Moderate pruning (e.g., $z_{\mathrm{prune}} \approx 2.0$) effectively balances memory efficiency and representation quality, making it a practical choice for long-term online learning scenarios. Additionally, we observe fairly similar performance on both datasets and expect that similar choices would perform well on other datasets.

Table 1: Comparison of classification accuracy and cluster purity for various $z_{\mathrm{prune}}$ thresholds in the memory consolidation step. Results represent the mean across three independently seeded trials $\pm$ one standard deviation.

| $z_{\mathrm{prune}}$ | ImageNet-100 | | | Places365-100 | | |
|---|---|---|---|---|---|---|
| | $A$ | $P$ | \|LTM\| | $A$ | $P$ | \|LTM\| |
| $\infty$ | $41.0 \pm 0.1$ | $30.0 \pm 0.4$ | $19472 \pm 813$ | $40.7 \pm 0.0$ | $27.2 \pm 0.3$ | $32152 \pm 1587$ |
| 2.25 | $40.5 \pm 0.5$ | $30.0 \pm 0.5$ | $18377 \pm 584$ | $41.0 \pm 0.1$ | $27.5 \pm 0.1$ | $29576 \pm 831$ |
| 2.0 | $41.3 \pm 0.2$ | $29.8 \pm 0.6$ | $15845 \pm 868$ | $41.3 \pm 0.2$ | $27.6 \pm 0.3$ | $27554 \pm 601$ |
| 1.75 | $38.1 \pm 0.6$ | $28.8 \pm 0.1$ | $10451 \pm 571$ | $37.9 \pm 0.6$ | $24.3 \pm 0.7$ | $9490 \pm 1547$ |

## 5  Discussion

This work highlights the unique challenges of Online Unsupervised Continual Learning (O-UCL). We examine the commonly held "single-pass" criteria which is common among online learning works and argue that it should not be a fundamental assumption of online learning. We further propose a new challenge which may be more relevant to an online learner, emphasizing the impact of short-term biases that hinder learning and cause what we call *catastrophic non-learning*. Additionally, we examine the unique challenges which the O-UCL environment presents in terms of building a memory to be used for replay. Arguing that such a memory in the O-UCL setting should not have a fixed memory cap, but should instead focus on efficient and balanced growth, despite the lack of labels, phase identities, and even when presented with imbalanced streams.

To address these challenges, we proposed a dual-memory framework that integrates a short-term memory (STM) for high-throughput learning and a long-term memory (LTM) that dynamically selects and retains novel examples based on evolving latent space similarities.

Our experimental results demonstrate that traditional or single-memory approaches fail to simultaneously mitigate both *catastrophic non-learning* from biased streams and *catastrophic forgetting* due to long-term shifts. In contrast, the dual-memory system significantly improves representation retention and adaptation, maintaining significantly improved classification and clustering performance across challenging ImageNet-100 and Places365-100 streams.

While our work demonstrates promising empirical results, we have not provided a corresponding theoretical analysis or justification of the proposed method. This omission stems from the inherently unconstrained nature of the O-UCL problem, which makes only minimal assumptions regarding key factors such as the independence and identical distribution (IID) of the data, the number of classes, or the total amount of available data. These relaxed conditions render rigorous theoretical treatment particularly challenging. Nevertheless, developing such analyses represents an important and promising direction for future research.

Our proposed approach is also memory and computationally efficient, retaining only a small fraction of the training stream while maintaining competitive performance. For ImageNet-100, the long-term memory (LTM) stored an average of $15,845 \pm 868$ images, corresponding to approximately 14.7% of the total training set, with an overall storage footprint (including short-term memory, STM) of $\sim 840$ MB. For Places365-100, the LTM stored $27,554 \pm 601$ images, or roughly 6.8% of the training data, requiring $\sim 1.35$ GB of storage. The memory systems are also implemented as parallel workers, allowing for the augmentations of memory images to be done in parallel with the forward pass and other model updates, ensuring the system is able to compute between 3-6 iterations per second on a single V100. These results highlight that our approach achieves substantial compression of the training stream while keeping memory usage within practical bounds. Despite the potential for our proposed approach to scale further, this work was done with limited computational resources and scaling experiments to the full ImageNet1k or Places365 datasets would be computationally prohibitive and so is left for future work.

These findings underscore the necessity of addressing both catastrophic forgetting and non-learning in real-world continual learning. Future work could explore further optimizations in adaptive memory selection, self-supervised objectives tailored for O-UCL, and scalable memory consolidation strategies to enhance online unsupervised representation learning.

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

Table 2: Hyperparameters used in our experiments for ImageNet-100 and Places365-100.

| Parameter | ImageNet-100 | Places365-100 | Description |
|---|---|---|---|
| **Training Parameters** | | | |
| Backbone | ResNet-18 | ResNet-18 | Architecture used for feature extraction. |
| Optimizer | SGD | SGD | Optimization algorithm used for training. |
| Learning Rate ($lr$) | 0.6 | 0.6 | Initial learning rate for encoder training. |
| Momentum | 0.9 | 0.9 | Momentum parameter for SGD. |
| Weight Decay | $1 \times 10^{-6}$ | $1 \times 10^{-6}$ | Regularization strength for weight decay. |
| Batch Size | 512 | 512 | Number of samples per training batch. |
| $B_{\mathrm{stream}}$ | 64 | 64 | |
| **STM Parameters** | | | |
| STM Size ($M_{\mathrm{stm}}$) | 2048 | 2048 | Maximum number of examples stored in short-term memory. |
| K | 10 | 10 | Number of examples to select for STM each iteration. |
| **LTM Parameters** | | | |
| $\beta$ | 90 | 87.5 | Sensitivity for calculating adaptive novelty threshold $d_{\mathrm{thresh}}$ |
| LTM Init Size ($M_{\mathrm{init}}$) | 512 | 512 | Number of initial examples stored in LTM. |
| Update Interval ($u$) | 500 | 500 | Frequency (in steps) at which LTM updates its stored embeddings. |
| Pruning Threshold ($Z_{\mathrm{prune}}$) | 2.0 | 2.0 | Z-score threshold for memory consolidation pruning. |
| Partitions ($P$) | 10 | 10 | Number of partitions for age-based pruning in LTM. |
| **Evaluation Parameters** | | | |
| KNN | 50 | 50 | Number of nearest neighbors for KNN. |
| KMeans | 2*N | 2*N | Number of clusters for KMeans where N is classes seen so far. |

Table 3: Dual Memory System Augmentation Hyper-Parameters

| Symbol | ImageNet-100 | Places365-100 |
|---|---|---|
| Color Jitter | 0.8 | 0.8 |
| Brightness | 0.8 | 0.8 |
| Contrast | 0.8 | 0.8 |
| Saturation | 0.8 | 0.8 |
| Hue | 0.2 | 0.8 |
| Horizontal Flip | 0.5 | 0.5 |
| Vertical Flip | 0.0 | 0.0 |
| Random Gray Scale | 0.2 | 0.2 |
| Gaussian Blur | 0.5 | 0.5 |
| Kernel Size | 5 | 12 |
| $\sigma_1$ | 0.1 | 0.1 |
| $\sigma_2$ | 2.0 | 2.0 |
| Crop Min | 0.08 | 0.08 |
| Crop Max | 1.0 | 1.0 |

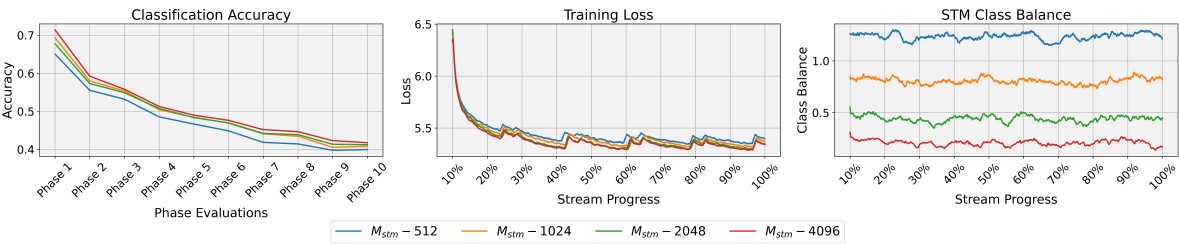

Figure 10: Classification accuracy (left), training loss (center), and STM class balance (right) for the dual-memory system with varying STM capacities on the ImageNet-100 datastream.

# A   Additional Dual-Memory System Analysis

## A.1   Impact of STM Size

In this subsection, we examine the impact of the short-term memory (STM) capacity on the performance and learning dynamics of the dual-memory system. Figure 10 presents the classification accuracy (left), training loss (center), and STM class balance (right) for four different values of $M_{\text{STM}}$, which denotes the size of the STM.

Analyzing the classification accuracy, we observe relatively minor differences across the different STM capacities, highlighting the efficiency of the STM in facilitating learning even with minimal memory usage. However, a slight decline in performance is noticeable when the STM size is reduced to 512, suggesting a lower bound beyond which performance begins to degrade. Additionally, the training loss exhibits a clear decreasing trend as the STM size increases. This trend is expected, as a larger STM provides a more diverse and representative set of samples, leading to a more accurate estimation of the true gradient for the current phase's classes.

Examining the STM class balance in the right subfigure, we observe that as the STM size decreases, class balance improves. This indicates that smaller STM capacities introduce greater bias in the batch samples drawn from memory, thereby making the learning task more challenging. These results underscore the importance of maintaining an appropriately sized STM to ensure a representative set of stored examples. However, they also suggest that effective learning can be achieved with relatively modest STM sizes, demonstrating the efficiency of the dual-memory approach in managing memory constraints.

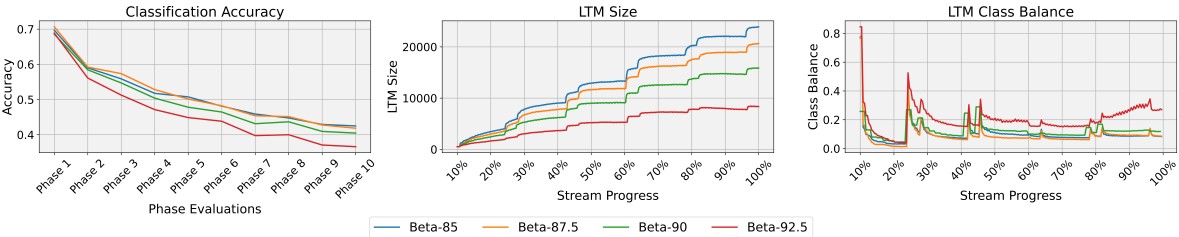

Figure 11: Classification accuracy (left), LTM size (middle), and LTM class balance (right) for the dual-memory system with varying novelty sensitivity values $\beta$ on the ImageNet-100 datastream.

## A.2 Impact of $\beta$

In this subsection, we analyze the impact of the long-term memory (LTM) novelty threshold sensitivity parameter, $\beta$. Figure 11 presents the classification accuracy (left), LTM size (center), and LTM class balance (right) for the dual-memory system across four different values of $\beta$.

Examining the classification accuracy, we observe minimal differences among the three lower values of $\beta$, with only $\beta = 92.5$ exhibiting a noticeably lower performance. The middle plot, which illustrates the growth of the LTM, reveals a clear relationship between increasing values of $\beta$ and the overall size of the LTM over time. Notably, the step-function-like growth pattern remains consistent across all values of $\beta$, indicating that the model's desired behavior is robust to variations in this parameter.

Finally, considering the class balance (right), we observe that the models achieving the highest classification accuracy—corresponding to the lowest values of $\beta$—tend to exhibit the lowest class balance. This suggests that when the novelty threshold is overly restrictive, the LTM struggles to construct a representative set of exemplars, ultimately degrading overall classification performance.

## A.3 Dual-Memory System Memory Usage

The total memory consumption of the dual-memory system for storing images is given by the sum of the memory allocated to the short-term memory (STM) and the final size of the long-term memory (LTM) at the end of the stream. In the case of ImageNet-100, with a parameter setting of $\beta = 90$ and $z_{\mathrm{prune}} = 2.0$, the LTM stored an average of $15845 \pm 868$ images. The ImageNet-100 dataset comprises 100 classes, each containing 1350 images. The training subset consists of 80% of these images per class, resulting in a total of 1080 images per class and an overall training set size of 108,000 images. Consequently, the dual-memory system retained approximately 14.7% of the total training images.

Since these images are resized to $128 \times 128$ pixels and stored in `uint8` format, the storage requirement per image is approximately 48 KB. Given that the LTM stored 15,845 images and the STM maintained 2,048 images, the total storage requirement is approximately 840 MB.

Similarly, for Places365-100, the LTM stored an average of $27554 \pm 601$ images. The number of images per class in Places365 is not entirely balanced, but the training subset consists of approximately 404,000 images. As a result, the dual-memory system retained around 6.8% of the total training data throughout the stream. The storage requirement for these images amounts to approximately 1.35 GB.

## A.4 Novelty Detection Dynamics

Figure 12 presents the evolution of the novelty threshold and the number of detected novelties throughout the stream. Observing the novelty threshold over time, we note a steady increase during the first three phases. This trend arises because the model begins from a randomly initialized state and must progressively structure the latent space to better separate the incoming data. Upon reaching the fourth phase, a distinct pattern emerges: the novelty threshold momentarily drops at the introduction of each new phase before recovering to an approximately stable level.

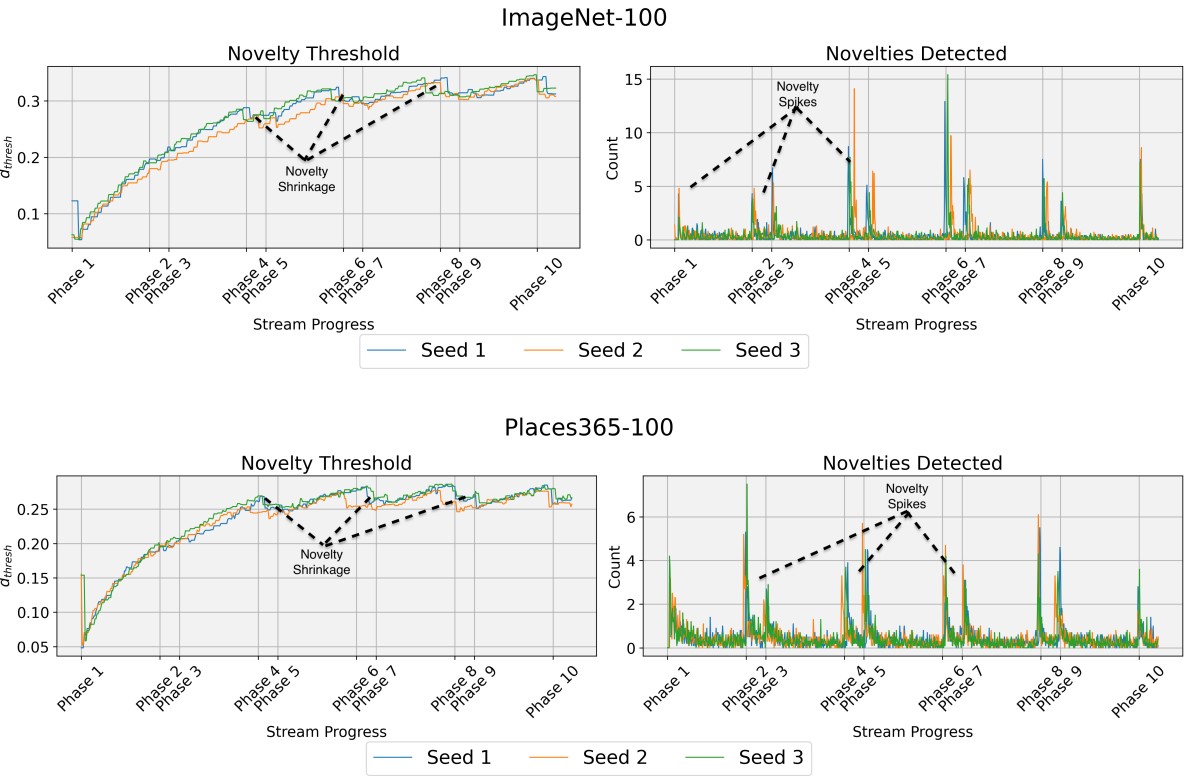

Figure 12: Novelty threshold $d_{\text{thresh}}$ (left) and novelties detected per step (right) for ImageNet-100 (top) and Places365-100 (bottom). Results are shown for three seeds on the phase-imbalanced streams from the primary results in the main paper. The annotations highlight the important trends in the novelty threshold (novelty shrinkage) and the count of novelties detected per step (novelty spikes) which result from changes in the underlying data distribution (new phases).

Examining the plot of detected novelties, we observe a pronounced spike at the onset of each new phase. This phenomenon occurs because the introduction of a new phase leads to a rapid expansion of the long-term memory (LTM) as novel examples are incorporated (novelty spikes). However, since this data represents a previously unseen distribution that the encoder has not yet adequately learned, the newly added representations are initially clustered more tightly in the latent space. This transient effect causes a temporary reduction in the novelty threshold $d_{\text{thresh}}$, as the average pairwise distances among data points decrease (novelty shrinkage). As the stream progresses and the model refines its representation of the new phase, the spread of the data increases. In the earlier phases, even the initial data undergoes significant restructuring as the model adapts, diminishing the relative impact of new phase introductions—since, at that stage, all incoming data is effectively novel.

## B  LTM Adaptation with Repeating Classes

In this section, we examine the long-term memory (LTM) system's ability to not only adapt to novel data introduced in a new phase but also to selectively disregard repeated data that has already been encountered. This characteristic is critical for memory efficiency, as the LTM's capacity is effectively unbounded. In scenarios where the phases of the data stream cycle through previously observed classes or data, an ideal model should be capable of recognizing and ignoring such redundant information in the context of LTM storage, while still leveraging it to improve classification performance through continued training on the streaming data.

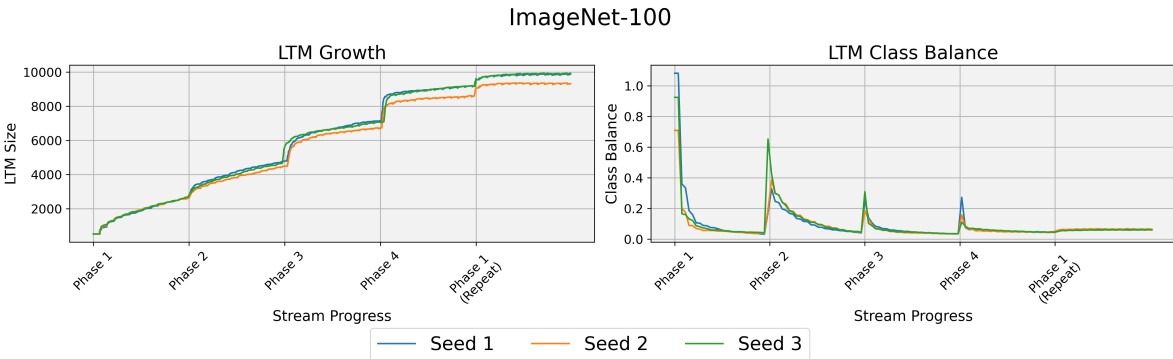

Figure 13: LTM growth (left) and LTM class balance (right) for a data stream with a repeated phase. The stream consists of four initial phases, each introducing 10 new classes, followed by a fifth phase that reintroduces the same classes from Phase 1, but in a shuffled order.

Figure 13 presents the LTM growth (left) and LTM class balance (right) for a data stream containing a repeated phase. The first four phases each consist of 10 classes, following the same setup as our primary experiments. However, instead of introducing a unique set of classes in the fifth phase, we reintroduce the same classes from Phase 1, albeit in a different order. Ideally, the LTM should recognize that this phase does not introduce novel information and, therefore, should exhibit minimal growth, with only minor changes in class balance.

The results indicate that while there is a modest increase in the LTM size at the onset of the repeated phase, the growth is substantially lower than what would be expected if the phase contained truly novel classes. Furthermore, the class balance remains largely unchanged, suggesting that the new additions primarily serve to correct class-wise imbalances in the LTM that arose during the first phase. This effect may stem from adjustments in the latent space as the encoder continued to refine its representations over the course of training. Since the LTM's selection of stored examples during the first phase was based on an undertrained encoder, some of those early storage decisions may have been suboptimal. Encountering the same classes again with a more refined encoder likely enabled improved decisions regarding which examples should be retained.

Additionally, due to the consolidation step, some of the images originally stored from Phase 1 were pruned from the LTM, meaning that the net increase in stored images from Phase 1 is slightly lower than the raw growth observed in the figure. Specifically, at the end of the first phase, the LTM contained 2,873 images. By the end of the stream, the total number of images stored for Phase 1 had increased to 3,328, representing a total increase of approximately 15%.

These findings demonstrate that the proposed LTM mechanism is capable of appropriately adapting to shifts in the underlying data distribution, effectively distinguishing between truly novel data and previously encountered information. This ability ensures both efficient memory usage and robust long-term adaptation in streaming environments.

## C   Phase-Wise Classification Performance

While the evaluations in this work primarily focus on a class-incremental setting, where a single output space expands as new classes are introduced, we also analyze the model's phase-wise performance. This evaluation measures the classification accuracy of each class independently and then averages the accuracy across all classes introduced within the same phase. Since the classification task becomes increasingly challenging as the stream progresses—due to the growing number of classes—we assess performance based on the final evaluation at the end of the stream. Given the highly imbalanced nature of the phase durations in the stream, it is expected that the model's ability to discriminate between classes introduced in shorter phases may be affected. To quantify this effect, we compare the phase-wise performance of the dual-memory model

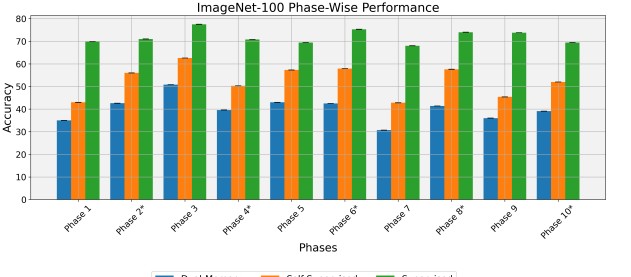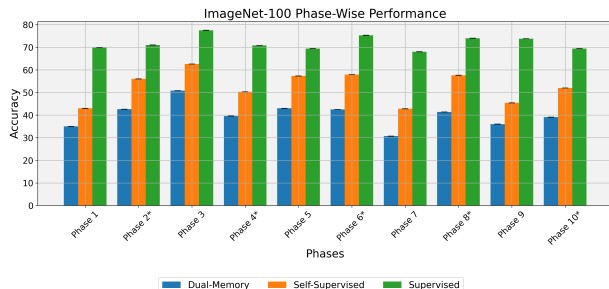

Figure 14: Phase-wise classification performance for ImageNet-100 (left) and Places365-100 (right). Phase-wise performance is measured as the average individual class performance at the final evaluation, averaged over the phase they were introduced. Phases marked with a * are to indicate their duration is 25% of the other phases length. The results represent the mean over three trials with the error representing ± one standard deviation.

against the two upper-bound baselines considered in Figure 8. We omit the lower-bound baselines, as both exhibit significantly reduced performance, making trend analysis less reliable.

Figure 14 presents the phase-wise classification performance for ImageNet-100 (left) and Places365-100 (right). Phases denoted with an asterisk correspond to significantly shorter durations. Despite the imbalanced length of the training phases, the test data used for the evaluation is balanced, showing that the model is able to reach similar performance on the classes which are observed for significantly fewer training steps. If the model's performance were biased toward longer phases, we would expect to observe a systematic reduction in accuracy for these asterisk-marked phases. The results indicate that while some variance exists across phases, the relative performance trends closely follow those of the upper bounds. This is particularly notable, as the upper-bound models were trained offline without exposure to phase imbalances. Thus, their performance trends more accurately reflect differences in the intrinsic difficulty of classifying the classes within each phase, rather than artifacts introduced by phase imbalances. These trends remain consistent across both the ImageNet-100 and Places365-100 data streams.

Overall, these findings demonstrate that the dual-memory system effectively mitigates the challenges associated with highly imbalanced class and phase distributions. The model's performance aligns with the inherent difficulty of the classification task rather than being disproportionately influenced by variations in data availability across phases.

## D  Further Online vs Offline Experiments

In this section, we extend the experiments presented in Section 2.1, which analyzed the importance of repeated exposure to training examples. Specifically, we examine the impact of both the number of training iterations and the total amount of unique training data on model performance.

Figure 15 presents a grid of experimental results where each row corresponds to a different number of training iterations, and each column represents a different percentage of the overall dataset used for training. The results in the first row for both ImageNet-100 and Places365-100 are identical to those in Figure 2. From these initial results, we progressively increase the number of training iterations as we move downward in the figure. Unlike the prior experiment, where the "online" model observed each training example exactly once, we now explore models trained with varying total amounts of data and varying numbers of training iterations.

Let $N$ denote the total number of unique examples available in the dataset, and let $T$ represent the total number of training iterations. The top-right cell in Figure 15 corresponds to an "online" setting where 100% of the dataset ($N$) is observed once over 200 iterations (i.e., a single pass over the data). Conversely, the

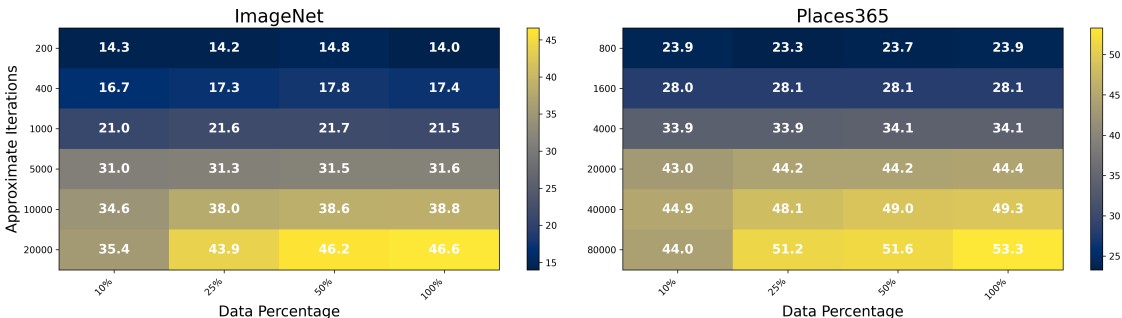

Figure 15: Classification accuracy for varying total number of training iterations and $N$ (total number of unique training examples). Results in the first row represent the equivalent of 10, 5, 2, 1 epochs and $\frac{N}{10}$, $\frac{N}{5}$, $\frac{N}{2}$, $\frac{N}{1}$ from left to right, ensuring an equivalent number of training iterations. Each subsequent row from top to bottom represents a multiple of the previous row in terms of iterations, which also increases the effective epochs (or repeats of individual images).

bottom-right cell corresponds to a training regime where the model undergoes 20,000 iterations, equivalent to approximately 100 epochs over the dataset.

Examining the results within each column, we observe that performance consistently improves as the number of training iterations $T$ increases, which aligns with expectations. However, for the first four rows, the total amount of data used (i.e., the percentage of $N$) has little impact on final accuracy. It is only in the last two rows—where the model undergoes a significantly larger number of iterations—that increasing the dataset size results in a measurable performance gain. Since the number of iterations per row is fixed, models trained on a smaller dataset size necessarily experience a higher number of repetitions per example. These results suggest that strong performance is primarily driven by the total number of training iterations $T$ and the dataset size $N$, rather than the frequency of repeated exposure to individual examples.

To further validate this conclusion, we compare the performance trends between ImageNet-100 and Places365-100. Despite similar upper bounds on classification accuracy, overall performance on Places365-100 is significantly higher. This discrepancy can be attributed to the larger dataset size of Places365-100, which allows for more unique training examples per iteration. Even when each example is seen only once, the larger $N$ provides sufficient training signal to match or exceed the performance of a model trained "offline" on a smaller or equivalent dataset size with an identical total number of iterations $T$. Consequently, we conclude that an "online" learner with access to a sufficiently large data stream can achieve comparable or superior performance to a model trained in a traditional "offline" setting, provided that $N$ and $T$ are sufficiently large.

# E   Comparison with MoCoV2 on Biased Streams

It is possible that the extreme reduction in performance induced by the bias streams in Section 2.1 are unique, or particularly substanital due to our use of the SimCLR Chen et al. (2020a) approach for self-supervised training. One possible alternative which may have intrinsic benefits is MoCo Chen et al. (2020b) which already utilizes a memory bank which may function somewhat like the STM in mitigating the impact of short-term bias and catastrophic non-learning. To examine this, we perform an experiment comparing a model trained with the SimCLR Chen et al. (2020a) approach to a model trained with the MoCoV2 Chen et al. (2020b) approach. For the SimCLR model we use the same hyper-parameters as described in Table 2 and for MoCoV2 we use the hyper-parameters for ImageNet described in the paper Chen et al. (2020b).

The results in Figure 16 show that the model trained with MoCoV2 actually has a slightly worse performance than SimCLR. We suspect that this is because while MoCo does have a memory bank of past *representations* it does not store raw examples. Because the model is struggling to learn at all from the highly biased samples, the stored representations are not useful and actually function as a form of noise, pushing the already highly

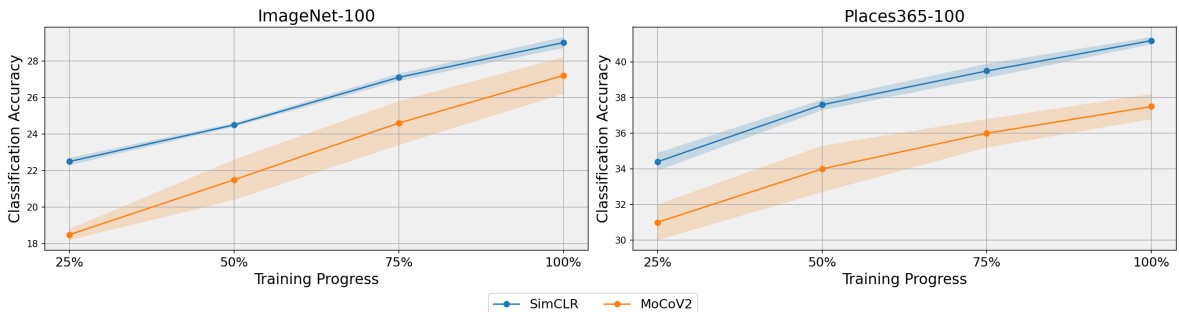

Figure 16: Classification accuracy of model trained with SimCLR and MoCoV2 methods on biased streams for 100 epochs. Each result represents the mean and standard deviation over six trials consisting of 3 dataset samples and two random seeds.

biased positive batch away from a large set of negatives which are not all that meaningful to begin with. We can see that MoCo closes the performance gap slightly later in training as the memory bank is filled with improved samples, but we can clearly see that it does not provide any performance benefit when compared with the SimCLR approach.

## F  Full Algorithm Psuedocode

**Algorithm 1** Python-Like psuedocode of the dual-memory system algorithm.

```python
# B - Total batch size
# M_stm - STM capacity
# K - number of new examples for STM per batch
# beta - novelty threshold senstivity
# M_init - initial size of LTM
# mu - encoder/d_thresh update frequency
# z_prune - memory consolidation sensitivity
# P - number of LTM partitions during consolidation
# encoder - The backbone with associated loss (e.g. simclr/resnet18)
stm = init_stm(M_stm, K)
ltm = init_ltm(encoder.deepcopy(), beta, m_init, mu, z_prune, P)

for t, X_stream in enumerate(S):
    # Calculate batch sizes per source
    M = stm.size() + ltm.size()
    B_m = B - X_stream.size()
    B_stm, B_ltm = (stm.size() / M) * B_m, (ltm.size() / M) * B_m

    # Create batch and perform training step
    X_stm, X_ltm = stm.sample(B_stm), ltm.sample(B_ltm)
    l_stream, l_stm, l_ltm = encoder.train_step(concatenate(X_stream, X_stm, X_ltm))

    # Update the STM and LTM
    stm_evicted = stm_update(stm, l_stream, X_stream)
    ltm_update(ltm, stm_evicted)
    if (t % mu) == 0:
        ltm.encoder = encoder.deepcopy()
        ltm_consolidate(ltm)

 def stm_update(stm, l_stream, X_stream):
     evicted_examples = stm.dequeue(stm.K) # pop K examples from the stm queue
     new_examples = X_stream[argsort(l_stream)[-stm.K:]] # top K examples
     stm.enqueue(new_examples)

     return stm

def ltm_update(ltm, stm_evicted):
    Z_stm = ltm.encoder(stm_evicted) # embed the stm examples
    D = 1 - (Z_stm @ ltm.Z.T) # pairwise distances between stm_evicted & LTM
    close_d = min(D, axis=1) # distance to nearest neighbor per example

    add_inds = where(close_d > ltm.d_thresh)
    ltm.X.add(stm_evicted[add_inds])
    ltm.Z.add(Z_stm[add_inds])

    return ltm

def ltm_consolidate(ltm):

    D = 1 - (ltm.Z @ ltm.Z.T) # calculate pairwise distances for each ltm embedding
    close_d = min(D, axis=1) # closest embedding for each example
    ltm.d_thresh = percentile(close_d, ltm.beta) # update d_thresh

    partitions = partition(ltm, ltm.P) # split the embeddings (Z) into P partitions

    for partition in partitions:
        # Calculate per partition distances and similarity score
        D_p = 1 - (partition.Z @ partition.Z.T)
        scores = z_score(D_p, ltm.z_prune)

        # Keep only the dissimilar examples
        keep_inds = where(scores > z_prune)
        partition.X = partition.X[keep_inds]
        partition.Z = partition.Z[keep_inds]

    return ltm
```

