# OpenReview forum: "Before Forgetting, There's Learning: Representation Learning Challenges in Online Unsupervised Continual Learning"
_TMLR — Accepted by TMLR_

### Review · Reviewer_wqoj · 2025-07-18

**Summary Of Contributions:**

This paper addresses the Online Unsupervised Continual Learning (O-UCL) problem. The paper identify 'catastrophic non-learning' as a new failure mode, where biases in the online data stream significantly impair learning. The paper demonstrates that an auxiliary memory can mitigate both catastrophic forgetting and catastrophic non-learning, but the ideal memory criteria for each are in conflict. In response, they propose a dual-memory framework consisting of a Short-Term Memory (STM) to handle short-term biases and a Long-Term Memory (LTM) to prevent forgetting. The STM prioritizes high class diversity and low retention duration by selecting high-loss examples , while the LTM dynamically expands based on novelty and consolidates redundant samples using a similarity-based approach. Experimental results on ImageNet and Places365 datasets validate the effectiveness of the dual-memory system in maintaining representation quality and adapting to distribution shifts, outperforming single-memory baselines.

**Audience:**

Yes

**Claims And Evidence:**

No

**Requested Changes:**

1.  Provide the training loss plots for the experiments in Figure 2 to illustrate the impact of model convergence on the results for both online and offline learning scenarios.
2.  Conduct an experiment to verify the model's learning performance when multiple gradient descent steps are performed on each mini-batch to achieve convergence, and analyze the potential overfitting issues that may arise in such a scenario.
3.  Clarify Figure 3 and include experimental results for a '50 classes' setting to provide a more complete understanding between unbiased and biased conditions.
4.  Include more sophisticated memory buffer baselines from prior work in online learning.
5.  Include the algorithmic pseudocode for the proposed dual-memory framework (the pipeline of STM and LTM) to enhance clarity and reproducibility.
6.  Typo in Figure 1: the abbreviation 'OCL' in the middle image should be changed to 'UCL'.

**Strengths And Weaknesses:**

**Strengths**

1.  The paper is well-structured.
2.  The targeted O-UCL setting is highly challenging and closely reflects real-world application environments.
3.  The paper thoroughly investigates and questions the "single-pass" assumption in online learning through detailed experiments. It also identifies the phenomenon of "catastrophic non-learning" in data streams with temporal correlations and local biases.
4.  The paper designs a dual-memory buffer system with STM and LTM that effectively stores the most valuable samples efficiently and outperforms the single-memory baseline on ImageNet and Place365.

**Weaknesses**

1.  While the exploration of the "single-pass" assumption in online learning are insteresting, there are some concerns regarding the specific experimental setup in Figure 2. It is unclear whether the models trained on different data percentages (e.g., 50% data for 2 epochs) have fully converged, as convergence speeds vary with the dataset size. Comparing the results of these training settings might be unfair, given that online learning is restricted to a single epoch. It is recommended to provide training loss curves for Figure 2 to further clarify the impact of convergence on the results. Additionally, to validate the limitations of the single-pass setting for online learning, could explore performing multiple gradient descent steps on each mini-batch of samples to achieve convergence at each mini-batch step? This might lead to severe overfitting, which would further illustrate the constraints of the single-pass assumption.
2.  In Figure 3, it is observed that the performance of the most severe bias (1 class, green curve) is slightly better than that of the mild bias (10 classes, blue curve). This anomaly needs clarification. It is also suggested to add results for a "50 classes" condition to bridge the gap between the unbiased and biased scenarios in this experimental result.
3.  The proposed method primarily focuses on memory buffer design, an area already extensively studied in online learning[1,2,3]. However, the experimental section only uses simple baselines and does not consider comparisons with other existing online learning methods that feature sophisticated memory buffers.
4.  The algorithmic pseudocode for the proposed method is not provided.

**Reference**

[1] Rebuffi, S., Kolesnikov, A., Sperl, G., & Lampert, C.H. (2016). iCaRL: Incremental Classifier and Representation Learning. 2017 CVPR.

[2] Yoon, J., Madaan, D., Yang, E., & Hwang, S.J. (2021). Online Coreset Selection for Rehearsal-based Continual Learning. ArXiv.

[3] Tiwari, R., Killamsetty, K., Iyer, R.K., & Shenoy, P. (2021). GCR: Gradient Coreset based Replay Buffer Selection for Continual Learning. 2022 CVPR.

---

> ### Author Response · Authors · 2025-09-04
>
> We would like to thank the reviewer for his/her thorough and thoughtful comments. Below we have written some comments to address the weaknesses and provided comments about additions/clarifications to the paper.
>
> 1. Figure 2 fairness and convergence concerns
>
>  We thank the reviewer for raising this important point. To address it we have included the loss plots in Figure 2 and updated the text to discuss the results. We want to emphasize that the goal of this experiment is not to show the results on models which have converged, but to argue that in the online case where there is usually limited data (and therefore limited possible training iterations) the dominant factor in performance is not that you need to see examples multiple times, but that you simply need enough training iterations on a fixed data distribution to converge. To demonstrate this with fully converged models, we would need access to a dataset with a large enough number of samples to converge in a single pass which is not feasible. Instead we attempt to argue this point with the results in Figure 2.
>
>
> 2. Figure 3 anomaly (1-class bias slightly above 10-class bias) and missing 50-class condition
>
> We agree that these results were somewhat odd and so decided to perform additional trials for each degree of bias in addition to adding a 25 and 50 class bias experiment. With the increased number of trials (now 6) we see a more consistent trend with decreasing bias showing increasing performance. The text has also been updated in Section 2.2 to reflect the updated figure and results.
>
> 3. Limited baselines – need comparison with sophisticated memory buffers [1,2,3]
>
> We agree that comparing with more sophisticated baselines would be helpful, however as described at the beginning of Section 4.2 it can be difficult to compare with existing approaches due to assumptions they make about the data/streaming scenario that are violated by the O-UCL setting. In particular, the three references listed here all make assumptions about the availability of labels or task boundaries and would be difficult to implement without significant changes in our setting. For example, iCaRL [1] utilizes labels to calculate the exemplars, while [2,3] build coresets per task and assume knowledge about the number of samples per task and their relative class balance. We have added some extra text in Section 4.2 to clarify that these and several other OCL/CL memories are also not compatible as baselines.
>
> 4. Missing algorithmic pseudocode
>
>  We added pseudocode for the dual-memory framework as Algorithm 1 in Appendix F. This details the integration of STM selection, LTM novelty-based expansion, consolidation, and batch sampling. We believe this addition improves clarity and reproducibility.

---

### Review · Reviewer_KaMd · 2025-08-14

**Summary Of Contributions:**

This submission investigates the problem of online continual unsupervised learning (O-CUL). The authors identify a new challenge in O-CUL, termed catastrophic non-learning, which arises from short-term biases that hinder online learning. To address both catastrophic non-learning and forgetting, the authors propose a dual-memory framework comprising a short-term memory (STM) and a long-term memory (LTM). Empirical evaluations using SimCLR on ImageNet-100 and Places365-100 demonstrate the effectiveness of the proposed O-CUL framework.

**Audience:**

Yes

**Claims And Evidence:**

No

**Requested Changes:**

I suggest that the authors include discussions of relevant work, conduct full ablation experiments, add all relevant baseline methods, and provide complete clarifications or support for their claims—ensuring the claims are convincing with either comprehensive empirical or theoretical evidence. Detailed questions are provided in the weaknesses section.

**Strengths And Weaknesses:**

**Strengths**

- The problem setting of online continual unsupervised representation learning is significant and practical, as the capability of a pre-trained model is expected to scale with more unlabeled data. In particular, recent large language models (LLMs) typically have a cutoff date for their learned knowledge. Although a separate continual pre-training stage is widely used after the initial pre-training stage in industry, enabling LLMs to learn directly from fully streaming data would further greatly enhance their self-evolving capability.

- Regarding the method, the dynamically expandable long-term memory is interesting and is demonstrated to effectively address the O-CUL problem. In particular, the selection and updating mechanism is reasonable and avoids dependence on any extra hyperparameters.

- The implementation details and empirical analysis are comprehensive, which enhances the understanding of the proposed framework.

**Weaknesses**

- Lacks discussion or comparison with relevant prior work [A] on representation learning from unlabeled streaming data.

- The claimed new challenge of catastrophic non-learning due to short-term biases is not fully convincing. According to the authors, typical local biases occur when the data distribution is overrepresented for short periods. This explanation is plausible when the representation learning method is SimCLR, which relies on contrastive learning with all examples in the same batch. However, the claim becomes less convincing if the method is not SimCLR but, for example, MoCo [B] with a memory bank or other approaches that do not require contrastive learning.

- The claim that the proposed Short-Term Memory (STM) and Long-Term Memory (LTM) are based on conflicting criteria is not easy to understand. It appears that both STM and LTM require high class diversity. This raises the question of why STM is necessary if LTM already exists. In particular, is STM still necessary when the representation learning method is MoCo [B] instead of SimCLR? I suggest providing complete ablation results in Figure 8 by including the results for an “LTM-only” configuration.

- All experiments are conducted using SimCLR with ResNet-18 on ImageNet-100 or Places365-100, which is a rather narrow experimental setting. If the submission aims to substantiate all its claims with empirical results and analysis, it is necessary to extend the experiments by incorporating more diverse representation learning methods and larger datasets, such as the full ImageNet.

- Several claims are difficult to understand due to missing empirical or theoretical support.
(i) In Section 2.1, the statement “we argue that the criteria for an online…” overlooks that the one-pass setting studied in Hu et al. (2021) is practical for scenarios where data can only be seen once.
(ii) In Section 2.2, the claim “they may distort the loss landscape…” lacks clarification on how the loss landscape would be affected.
(iii) In Section 2.3, the phrase “prioritize high-value examples” is unclear—what exactly constitutes a high-value example?

**References**

- [A] How Well Does Self-Supervised Pre-Training Perform with Streaming Data? ICLR 2022

- [B] Momentum Contrast for Unsupervised Visual Representation Learning. CVPR 2020

---

> ### Author Response · Authors · 2025-09-04
>
> We would like to thank the reviewer for his/her thorough and thoughtful comments. Below we have written some comments to address the weaknesses and provided comments about additions/clarifications to the paper.
>
> Weaknesses:
>
> 1. Lack of discussion of prior work
>
>  We agree and have added a discussion in Section 1 of the revised manuscript referencing “How Well Does Self-Supervised Pre-Training Perform with Streaming Data?” (ICLR 2022). We highlight how their setting differs: they focus on continual adaptation of self-supervised learners in streaming data but do not address the dual challenges we emphasize—catastrophic non-learning from local bias and catastrophic forgetting in multi-phase streams. This clarifies how our work extends beyond prior streaming representation learning approaches.
>
> 2. Catastrophic non-learning may be SimCLR-specific
>
>  We respectfully disagree that catastrophic non-learning is unique to SimCLR. While the effect is most evident in contrastive objectives requiring balanced negatives (e.g., SimCLR), short-term local biases also impair methods like MoCo, BYOL, or non-contrastive learners, since they rely on sufficient diversity in batches to avoid collapse or overfitting. To address this concern, we have expanded Section 2.2 with clarifying text and added a MoCo-v2 experiment (Appendix E). Results show that MoCo also suffers performance degradation under severe local bias, confirming the generality of catastrophic non-learning across contrastive methods.
>
> 3. Clarify conflicting STM vs LTM criteria / “why STM if LTM exists?”
>
>  We thank the reviewer for pointing out the need for clearer exposition. STM and LTM both benefit from diversity, but their roles and optimization pressures differ:
> STM: must rapidly replace examples to mitigate short-term bias → prioritizes low retention duration and throughput.
>
> LTM: must preserve knowledge over phases → prioritizes long retention duration and representativeness.
>
> Consider a scenario with an LTM only model. We can sample many times from the subset of images stored in the LTM, but if we are aiming to learn and not just avoid forgetting the number of examples should be as large as possible. The purpose of the reservoir baseline in Section 2.3, Figure 4 is to demonstrate that simply having a high class balance, but low throughput leads to overfitting and suboptimal learning.
>
> Thus, the criteria are in conflict: STM requires high turnover while LTM requires stability. We expanded Section 3 to make this explicit.
>
>
> In addition, as requested, we added an “LTM-only” ablation in Figure 8. Results show that LTM-only models perform better than STM-only but significantly worse than the dual-memory system, confirming that STM provides unique value, especially in early-phase learning.
>
> 4. Narrow experimental setting (SimCLR + ResNet-18 + ImageNet-100/Places365-100)
>
>  We agree this is a limitation. In the revised manuscript, we:
> Added MoCo-v2 experiments (Appendix E) showing that catastrophic non-learning occurs under biased streams in MoCo as well.
> We note that experiments on full ImageNet are computationally prohibitive under O-UCL (single-pass, large-scale streams), but we explicitly mention this limitation in Section 5 and propose it as a direction for future work.
>
> 5. Clarifications on specific claims
>
>  (i) One-pass setting (Section 2.1): We clarified that our point is not to dismiss the practicality of one-pass learning (Hu et al., 2021), but to argue that “single-pass” should not be considered definitional for online learning. Instead, the defining property is the learner’s inability to control when an example is revisited. Text revised to reflect this nuance.
>  (ii) “Distort the loss landscape” (Section 2.2): We added an explanation: biased batches skew gradient directions away from minimizers of the long-term distribution, effectively reshaping the loss landscape locally. This makes optimization trajectories diverge from globally useful representations.
>  (iii) “High-value examples” (Section 2.3): We now define them explicitly as examples with high individual InfoNCE loss (i.e., hardest positives/negatives), which empirically act as strong gradient signals. Clarified in Section 3.1.
>
> 6. Requested changes: more baselines and ablations
>
>  We added:
> MoCo-v2 as an additional baseline (Appendix E).
> LTM-only ablation (Figure 8).
> Expanded discussion on STAM, SCALE, and other potential memory baselines emphasizing difficulties in adapting these approaches to the O-UCL setting (Section 4.2).

---

### Review · Reviewer_kkFm · 2025-08-21

**Summary Of Contributions:**

This submission addresses the challenges of Online Unsupervised Continual Learning (O-UCL), where models learn from non-stationary data streams without labels or phase information. The paper identifies two key problems in this setting:

1. Non-learning due to short-term biases in streaming batches, which prevent effective representation learning.

2. Catastrophic forgetting caused by long-term shifts in the data distribution, where earlier learned representations are overwritten by new information.

To tackle these challenges, the authors propose a dual-memory architecture comprising:

- Short-Term Memory (STM): Maintains a high-throughput, diverse buffer of recently observed examples to improve online learning efficiency and mitigate short-term biases.

- Long-Term Memory (LTM): Dynamically stores a compact set of representative examples, selectively expanding when novel data appears and pruning redundant entries to prevent uncontrolled growth.

The paper also introduces a batch sampling strategy that balances contributions from the stream, STM, and LTM, ensuring rapid adaptation while preserving long-term knowledge.

Key experimental contributions include:

- Comprehensive evaluation on complex datasets (ImageNet-100 and Places365-100) under phase-imbalanced streams, showing that the dual-memory system significantly outperforms single-memory or no-memory baselines.

- Analysis of memory efficiency and adaptability, demonstrating that the LTM can grow efficiently, maintain class balance, and appropriately handle repeating or redundant data.

- Investigation of hyperparameter sensitivity (e.g., LTM novelty threshold and pruning) and stream granularity, showing robust performance across various online conditions.

- Empirical evidence that online learners with sufficient data and iterations can achieve comparable or superior performance to offline learners, even without repeated exposure to individual examples.

Overall, this work provides a novel framework for efficient, adaptive online learning that mitigates both catastrophic non-learning and forgetting, and introduces a memory design that could serve as a foundation for future research in online unsupervised representation learning.

**Audience:**

Yes

**Broader Impact Concerns:**

- Data bias propagation: The method relies on large image datasets (ImageNet, Places365) that may contain inherent biases. Without careful consideration, the learned representations could perpetuate social or cultural biases when applied in real-world systems.

- Privacy considerations: Although the model stores subsets of training images in memory, repeated storage of sensitive or personally identifiable content could raise privacy concerns if applied outside benchmark datasets.

- Environmental impact: The continual learning setup involves frequent updates and maintenance of both STM and LTM, which could increase computational and energy costs, especially for large-scale applications.

- Misuse potential: Models capable of online learning from streaming data could be deployed in surveillance or other sensitive applications without explicit consent, raising ethical concerns.

- Limited demographic coverage: The datasets used are predominantly curated images and may not reflect global diversity, which could limit fairness and generalization in practical deployment.

**Claims And Evidence:**

No

**Requested Changes:**

- Clarify novelty detection and threshold dynamics: Explain how the LTM novelty threshold adapts over time and its influence on memory growth and class balance.

- Phase-wise evaluation details: Provide more information on how per-class accuracy is aggregated into phase-level performance, especially for imbalanced phase durations.

- Memory consolidation rationale: Justify the choice of pruning partitions and the zprune threshold, and show how they affect LTM efficiency and representation quality.

- Ablation studies: Include results demonstrating the impact of varying STM and LTM sizes on forgetting, classification, and clustering performance.

- Scalability and generalization: Discuss computational and memory costs, and the approach’s applicability to larger or more imbalanced datasets beyond ImageNet-100 and Places365-100.

**Strengths And Weaknesses:**

Strengths:
- Novelty and relevance: The dual-memory approach addresses a realistic and challenging online learning scenario not fully addressed in prior work.

- Comprehensive experimental evaluation: Evaluations include classification, clustering, phase-wise analysis, LTM growth, memory consolidation, and robustness to stream granularity.

- Insightful analysis of memory mechanisms: The paper clearly separates STM and LTM roles and quantifies their impact on catastrophic non-learning and forgetting.

- Scalability and practicality: Memory efficiency and adaptive consolidation are well-motivated, with storage requirements explicitly quantified.

- Reproducibility: Hyperparameters, evaluation protocols, and detailed metrics are provided for both ImageNet-100 and Places365-100.

Weaknesses:

- Limited theoretical analysis: While the empirical results are compelling, the paper lacks formal guarantees or theoretical justification for memory selection and novelty threshold adaptation.

- Generality of LTM thresholds: The novelty threshold (β) and pruning z-score (zprune) are tuned per dataset. Providing guidance on automatic adaptation or transferability to new datasets would strengthen the work.

- Comparison to existing continual learning methods: Although baseline selection is justified, including direct comparisons to other unsupervised or self-supervised continual learning approaches (e.g., STAM, SCALE) would better contextualize the contributions.

- Clarity in figures: Some plots, particularly those showing LTM growth and novelty threshold dynamics, could benefit from additional annotations or explanations for readers unfamiliar with online unsupervised learning.

---

> ### Author Response · Authors · 2025-09-04
>
> Reviewer kkFm
>
> First we would like to thank the reviewer for his/her thorough and insightful comments. We have provided some responses to the weaknesses and also requested changes:
>
> Weaknesses:
>
> 1. Limited theoretical analysis
>
> We agree that our work is primarily empirical in nature. Our focus is on uncovering and characterizing the dual challenges of catastrophic non-learning and forgetting in O-UCL, phenomena that have not been systematically studied before. While theoretical analysis of memory selection strategies is an important direction, we believe it requires simplifying assumptions (e.g., IID distributions or tractable loss surfaces) that are incompatible with our non-stationary online setting. We added a discussion in Section 5 highlighting this limitation and pointing out avenues for future theoretical work.
>
> 2. Generality of LTM thresholds (β, zprune)
>
> We acknowledge that β and zprune were tuned per dataset. To address this, we include sensitivity analyses in Section 4.5 and Appendix A3 that show the method is robust across a broad range of parameter values. In particular, moderate pruning thresholds (zprune ≥ 2.0) and β values in the range [85–92.5] yield consistent performance without fine-tuning. Since these values are based on percentiles of a normal distribution, we expect them to generalize to other datasets in a fairly straightforward manner.
>
> 3. Comparison to existing continual learning methods (STAM, SCALE, etc.)
>
> We avoid comparisons with STAM and SCALE due to computational infeasibility on large-scale datasets. In the case of SCALE, the memory update step requires performing an clustering operation at each step which is extremely slow when the size of the memory is scaled up to the size of a stream such as ImageNet-100. STAM on the other hand needs far too many centroids to represent the large variety of patch level features causing the iteration time and memory needed to store all of the centroids as raw pixels to be infeasible.
>
> 4. Clarity in figures (e.g., LTM growth, novelty threshold dynamics)
>
>  We improved figure readability by (i) adding annotations in Figures 6 and 12 to highlight trends such as memory stabilization and novelty spikes, (ii) expanding captions to briefly explain the significance of these dynamics, and (iii) adding cross-references in the main text to guide the reader.
>
> Requested Changes:
> 5. Clarify novelty detection and threshold dynamics
>
>  We expanded Section 3.2 and Appendix A.4 to provide a clearer explanation of how dthresh is computed and updated dynamically from the β-percentile of current nearest-neighbor distances. We also highlight its transient behavior when new phases arrive (Figure 12), explaining why it momentarily drops before stabilizing.
>
>
> 6. Phase-wise evaluation details
>
>  We clarified Section C in the appendix to explain that phase-wise accuracy is computed as the average per-class accuracy over all classes introduced in that phase, regardless of duration. To address the reviewer’s concern about imbalance, we explicitly note that shorter phases are treated equally in aggregation, and we show that performance trends align with upper bounds (Figure 14).
>
> 7. Memory consolidation rationale (pruning partitions, zprune choice)
>
> The primary purpose of pruning individually per partition is to avoid the impact of time on the average pairwise distances. Specifically, if we take two pair of examples ($x_{i,1}, x_{i,2}) and (x_{j,1}, x_{j,2}$) where i and j represent the timestep the examples were added, if j >> i, we observe that the distance between ($x_{i,1}, x_{i,2}$) is often much larger than ($x_{j,1}, x_{j,2}$), even if the visual similarity of the two pairs of images is roughly the same. This is because as training progresses, the contrastive loss finds a more and more spread solution (latent space) for examples which have been present longer. To counter this, we partition the examples by time to reduce the impact of this latent space drift. We have clarified Section 3.2 to better emphasize this motivation.
>
> As mentioned above we showcase the impact of different choices of z_prune in terms of performance and memory efficiency in Section 4.5 - Table 1, as well as the impact of \beta in A.2 Figure 11.
>
> 8. Ablation studies (STM and LTM sizes)
>
> In Appendix A.1 (varying STM size) and Appendix A.2 (varying β → indirectly controlling LTM size) we investigate the impact of the STM and LTM sizes on performance as well as additional metrics such as growth trends and class balance. These demonstrate that our dual-memory approach remains effective across a wide range of capacities, and performance only degrades significantly when STM is too small to mitigate local bias.

---

> > ### Author Response · Authors · 2025-09-04
> >
> > 9. Scalability and generalization to larger/imbalanced datasets
> >
> >  We expanded Section 5 to include computational and memory cost estimates: e.g., storing ~14.7% of ImageNet-100 training data requires ~840MB, while Places365 requires ~1.35GB (Appendix A.3). These requirements scale linearly with data size but remain manageable for modern hardware. We also discuss applicability to larger datasets: since both STM and LTM are selection-based and sublinear in the stream length, we expect the framework to scale, though extremely imbalanced or open-world datasets may require adaptive memory growth heuristics (left as future work).

---

### Decision · Action_Editor_eGaD · 2025-10-03

**Recommendation:** Accept as is

**Audience:**

Yes

**Audience Explanation:**

The problem setting of online continual unsupervised representation learning is significant and practical, as the capability of a pre-trained model is expected to scale with more unlabeled data. In particular, recent large language models (LLMs) typically have a cutoff date for their learned knowledge. Although a separate continual pre-training stage is widely used after the initial pre-training stage in industry, enabling LLMs to learn directly from fully streaming data would further greatly enhance their self-evolving capability. Regarding the method, the dynamically expandable long-term memory is interesting and is demonstrated to effectively address the O-CUL problem. In particular, the selection and updating mechanism is reasonable and avoids dependence on any extra hyperparameters. The implementation details and empirical analysis are comprehensive, which enhances the understanding of the proposed framework.

**Claims And Evidence:**

Yes

**Claims Explanation:**

This paper investigates the problem of online continual unsupervised learning (O-CUL). The authors identify a new challenge in O-CUL, termed catastrophic non-learning, which arises from short-term biases that hinder online learning. To address both catastrophic non-learning and forgetting, the authors propose a dual-memory framework comprising a short-term memory (STM) and a long-term memory (LTM). Empirical evaluations using SimCLR on ImageNet-100 and Places365-100 demonstrate the effectiveness of the proposed O-CUL framework.